# SYMPHONY: SYMMETRY-EQUIVARIANT POINT-CENTERED SPHERICAL HARMONICS FOR 3D MOLECULE GENERATION

**Ameya Daigavane[1], Song Kim[1], Mario Geiger[2]**[*]**, Tess Smidt[1]**
{ameyad,songk}@mit.edu, geiger.mario@gmail.com, tsmidt@mit.edu
[1]Massachusetts Institute of Technology    [2]NVIDIA

## ABSTRACT

We present Symphony, an $E(3)$-equivariant autoregressive generative model for 3D molecular geometries that iteratively builds a molecule from molecular fragments. Existing autoregressive models such as G-SchNet (Gebauer et al., 2019) and G-SphereNet (Luo & Ji, 2022) for molecules utilize rotationally invariant features to respect the 3D symmetries of molecules. In contrast, Symphony uses message-passing with higher-degree $E(3)$-equivariant features. This allows a novel representation of probability distributions via spherical harmonic signals to efficiently model the 3D geometry of molecules. We show that Symphony is able to accurately generate small molecules from the QM9 dataset, outperforming existing autoregressive models and approaching the performance of diffusion models.

## 1 INTRODUCTION

In silico generation of atomic systems with diverse geometries and desirable properties is important to many areas including fundamental science, materials design, and drug discovery (Anstine & Isayev, 2023). The direct enumeration and validation of all possible 3D structures is computationally infeasible and does not in itself lead to useful representations of atomic systems for guiding understanding or design. Machine learning methods offer a promising avenue to explore this landscape by learning to generate 3D molecular structures.

Effective generative models of atomic systems must learn to represent and produce highly-correlated geometries that represent chemically valid and energetically favorable configurations. To do this, they must overcome several challenges:

- The validity of an atomic system is ultimately determined by quantum mechanics. Generative models of atomic systems are trained on 3D structures relaxed through computationally-intensive quantum mechanical calculations. These models must learn to adhere to chemical rules, generating stable molecular structures based solely on examples.

- The stability of atomic systems hinges on the precise placement of individual atoms. The omission or misplacement of a single atom can result in significant property changes and instability.

- Atomic systems have inherent symmetries. Atoms of the same element are indistinguishable, so there is no consistent way to order atoms within an atomic system. Additionally, atomic systems lack unique coordinate systems (global symmetry) and recurring geometric patterns occur in a variety of locations and orientations (local symmetry).

Taking these challenges into consideration, the majority of generative models for atomic systems operate on point geometries and use permutation and Euclidean symmetry-invariant or equivariant methods. Thus far, two approaches have been emerged as effective for directly generating general 3D geometries of molecular systems: autoregressive models (Gebauer et al., 2019; 2022; Luo & Ji, 2022; Simm et al., 2020; 2021) and diffusion models (Hoogeboom et al., 2022).

In this work, we introduce Symphony, an autoregressive generative model that uses higher-degree equivariant features and spherical harmonic projections to build molecules while respecting the

---

[*]Work performed when at Massachusetts Institute of Technology.

$E(3)$ symmetries of molecular fragments. Similar to other autoregressive models, Symphony builds molecules sequentially by predicting and sampling atom types and locations of new atoms based on conditional probability distributions informed by previously placed atoms. However, Symphony stands out by using spherical harmonic projections to parameterize the distribution of new atom locations. This approach enables predictions to be made using features from a single 'focus' atom, which serves as the chosen origin for that step of the generation process. It allows for the simultaneous prediction of the radial and angular distribution of possible atomic positions in a direct manner without needing to use additional atoms.

To test our proposed architecture, we apply Symphony to the QM9 dataset and show that it outperforms previous autoregressive models and is competitive with existing diffusion models on a variety of metrics. We additionally introduce a metric based on the bispectrum for assessing the angular accuracy of matching generated local environments to similar environments in training sets. Finally, we demonstrate that Symphony can generate valid molecules at a high success rate, even when conditioned on unseen molecular fragments.

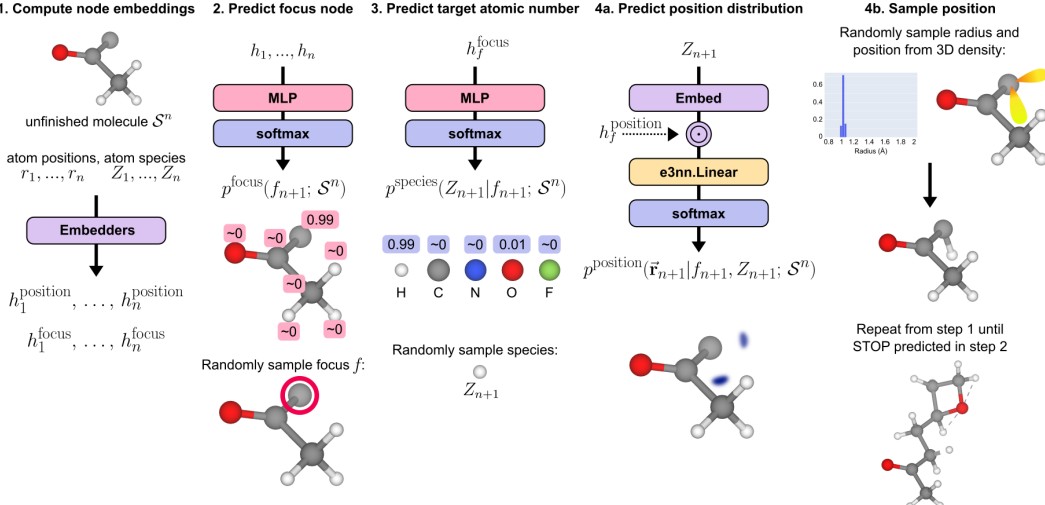

Figure 1: One iteration of the Symphony molecular generation process, in which one atom is sampled given the positions and atom types of an unfinished molecular fragment $\mathcal{S}^n$. The complete molecule after all iterations is shown in the bottom right of the figure.

## 2 BACKGROUND

$E(3)$-**Equivariant Features**: We say a $E(3)$-equivariant feature $z \in \mathbb{R}^{2l+1}$ transforms as the irreducible representation $l$ under rotation $\mathbf{R}$ and translation $\mathbf{T}$:

$$z \xrightarrow{\mathbf{R},\mathbf{T}} D^l(\mathbf{R})^T z$$

where $D^l$ is the irreducible representation of $SO(3)$ of degree $2l + 1$. $D^l(\mathbf{R}) \in \mathbb{R}^{(2l+1)\times(2l+1)}$ is referred to as the Wigner D-matrix of the rotation $\mathbf{R}$. As $D^0(\mathbf{R}) = 1$ and $D^1(\mathbf{R}) = \mathbf{R}$, invariant 'scalar' features correspond to degree $l = 0$ features, while 'vector' features correspond to $l = 1$ features. Note that these features are invariant under translation $\mathbf{T}$.

**Spherical Harmonics**: The real spherical harmonics $Y_{l,m}(\theta, \phi)$ are a set of real-valued orthogonal functions defined on the sphere $S^2$, indexed by two integers $l$ and $m$ such that $l \geq 0, |m| \leq l$. Here $\theta$ and $\phi$ come from the notation for spherical coordinates, where $r$ is the distance from an origin, $\theta \in [0, \pi]$ is the polar angle and $\phi \in [0, 2\pi)$ is the azimuthal angle. The relation between Cartesian and spherical coordinates is given by: $x = r \sin\theta \cos\phi, y = r \sin\theta \sin\phi, z = r \cos\theta$.

$l$ corresponds to an angular frequency: the higher the $l$, the more rapidly $Y_{l,m}$ changes over $S^2$. This can intuitively be seen by looking at the functional form of the spherical harmonics. In their Cartesian form, the spherical harmonics are proportional to simple polynomials. In one common choice of basis, $l = 0$ is proportional to 1, $l = 1$ is proportional to $(x, y, z)$ and $l = 2$ is proportional to $(xy, yz, 2z^2 - x^2 - y^2, zx, x^2 - y^2)$, as seen in Figure 3D-F.

One important property of the spherical harmonics is that they can be used to create $E(3)$-*equivariant* features. Let $Y_l(\theta, \phi) = [Y_{l,-l}(\theta, \phi), \ldots, Y_{l,l}(\theta, \phi)] \in \mathbb{R}^{2l+1}$ represent the collection of all spherical harmonics with the same $l$. Then, $Y_l(\theta, \phi)$ transforms as an $E(3)$-equivariant feature of degree $l$ under rotation: $Y_l(\mathbf{R}(\theta, \phi)) = D^l(\mathbf{R})^T Y_l(\theta, \phi)$, where $\mathbf{R}$ is an arbitrary rotation, and $(\theta, \phi)$ is interpreted as the coordinates of a point on $S^2$.

The second important property of the spherical harmonics that we employ is the fact that they form an *orthonormal basis* for functions on the sphere $S^2$. Thus, for any function $f : S^2 \to \mathbb{R}$, we can express $f$ as a linear combination of the $Y_{l,m}$. Formally, there exists unique coefficients $c_l \in \mathbb{R}^{2l+1}$ for each $l \in \mathbb{N}$, such that $f(\theta, \phi) = \sum_{l=0}^{\infty} c_l^T Y_l(\theta, \phi)$. We term these coefficients $c_l$ as the spherical harmonic coefficients of $f$ as they are obtained by projecting $f$ onto the spherical harmonics.

## 3 METHODS

We first describe Symphony, our autoregressive model for 3D molecular structures, with a comparison to prior work in Section 3.6.

### 3.1 BUILDING MOLECULES VIA SEQUENCES OF FRAGMENTS

First, we create sequences of fragments using molecules from the training set via CREATEFRAGMENTSEQUENCE. Given a molecule $\mathcal{M}$ and random seed $r$, CREATEFRAGMENTSEQUENCE constructs a sequence of increasingly larger fragments $\{\mathcal{S}^1, \ldots \mathcal{S}^{|\mathcal{M}|}\}$ such that $|\mathcal{S}^n| = n$ for all $n \in \{1, \ldots, |\mathcal{M}|\}$ and $\mathcal{S}^{|\mathcal{M}|} = \mathcal{M}$ exactly. Of course, there are many ways to create such sequences of fragments; CREATEFRAGMENTSEQUENCE simply builds a molecule via a minimum spanning tree.

Symphony attempts to recreate this sequence step-by-step, learning the (probabilistic) mapping $\mathcal{S}^n \to \mathcal{S}^{n+1}$. In particular, we ask Symphony to predict the focus node $f_{n+1}$, the target atomic number $Z_{n+1}$ and the target position $\vec{r}_{n+1}$, providing feedback at every step.

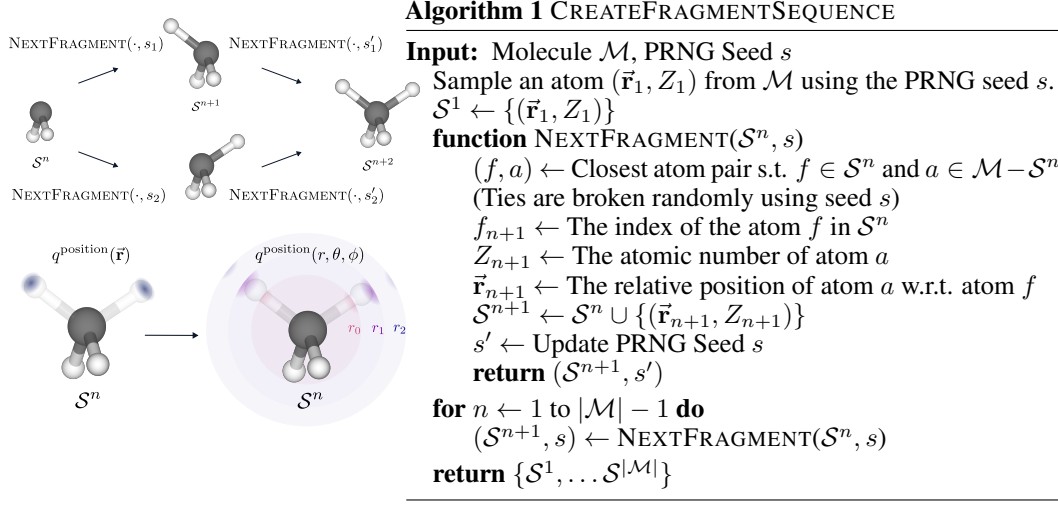

**Algorithm 1** CREATEFRAGMENTSEQUENCE

**Input:** Molecule $\mathcal{M}$, PRNG Seed $s$
Sample an atom $(\vec{r}_1, Z_1)$ from $\mathcal{M}$ using the PRNG seed $s$.
$\mathcal{S}^1 \leftarrow \{(\vec{r}_1, Z_1)\}$
**function** NEXTFRAGMENT$(\mathcal{S}^n, s)$
    $(f, a) \leftarrow$ Closest atom pair s.t. $f \in \mathcal{S}^n$ and $a \in \mathcal{M} - \mathcal{S}^n$
    (Ties are broken randomly using seed $s$)
    $f_{n+1} \leftarrow$ The index of the atom $f$ in $\mathcal{S}^n$
    $Z_{n+1} \leftarrow$ The atomic number of atom $a$
    $\vec{r}_{n+1} \leftarrow$ The relative position of atom $a$ w.r.t. atom $f$
    $\mathcal{S}^{n+1} \leftarrow \mathcal{S}^n \cup \{(\vec{r}_{n+1}, Z_{n+1})\}$
    $s' \leftarrow$ Update PRNG Seed $s$
    **return** $(\mathcal{S}^{n+1}, s')$
**for** $n \leftarrow 1$ to $|\mathcal{M}| - 1$ **do**
    $(\mathcal{S}^{n+1}, s) \leftarrow$ NEXTFRAGMENT$(\mathcal{S}^n, s)$
**return** $\{\mathcal{S}^1, \ldots \mathcal{S}^{|\mathcal{M}|}\}$

Figure 2: (Top) Fragments from CREATEFRAGMENTSEQUENCE applied to methane (CH4). From $\mathcal{S}^n$, there are thus two valid positions to place the next H atom around the focus $f_{n+1}$. (Bottom-Left) The true probability distribution $q^{\text{position}}(\vec{r})$ is smoothly projected onto (Bottom-Right) radial shells of spherical signals according to Equation 7. All the radial shells show the same angular distribution, but the shell corresponding to $r_1$ is the most probable.

### 3.2 HANDLING THE SYMMETRIES OF FRAGMENTS

Here, we highlight several challenges that arise because $\mathcal{S}^n$ must be treated as an unordered set of atoms that live in 3D space. In particular, let $\mathbf{R}\mathcal{S}^n + \mathbf{T} = \{(\mathbf{R}\vec{r}_1 + \mathbf{T}, Z_1), \ldots, (\mathbf{R}\vec{r}_n + \mathbf{T}, Z_n)\}$ be the description of the same set of atoms in $\mathcal{S}^n$ but in a coordinate frame rotated

by $\mathbf{R}^{-1}$ and translated by $\mathbf{T}^{-1}$. Similarly, let $\pi$ be any permutation of $\{1, \ldots, n\}$ and $\pi \mathcal{S}^n = \{(\vec{\mathbf{r}}_{\pi(1)}, Z_{\pi(1)}), \ldots, (\vec{\mathbf{r}}_{\pi(n)}, Z_{\pi(n)})\}$. Fundamentally, $\mathbf{R}\mathcal{S}^n + \mathbf{T}$, $\mathcal{S}^n$ and $\pi\mathcal{S}^n$ all represent the same set of atoms. Thus, we would like Symphony to naturally accommodate the symmetries of fragment $\mathcal{S}^n$, under the group $E(3)$ of Euclidean transformations consisting of all rotations $\mathbf{R}$ and translations $\mathbf{T}$, and the group of all permutations of constituent atoms. Formally, we wish to have:

- **Property (1)**: The focus distribution $p^{\text{focus}}$ and the target species distribution $p^{\text{species}}$ should be $E(3)$-*invariant*:

$$p^{\text{focus}}(f_{n+1}; \mathbf{R}\mathcal{S}^n + \mathbf{T}) = p^{\text{focus}}(f_{n+1}; \mathcal{S}^n) \tag{1}$$

$$p^{\text{species}}(Z_{n+1} \mid f_{n+1}; \mathbf{R}\mathcal{S}^n + \mathbf{T}) = p^{\text{species}}(Z_{n+1} \mid f_{n+1}; \mathcal{S}^n) \tag{2}$$

- **Property (2)**: The target position distribution $p^{\text{position}}$ should be $E(3)$-*equivariant*:

$$p^{\text{position}}(\mathbf{R}\vec{\mathbf{r}}_{n+1} + \mathbf{T} \mid f_{n+1}, Z_{n+1}; \mathbf{R}\mathcal{S}^n + \mathbf{T}) = p^{\text{position}}(\vec{\mathbf{r}}_{n+1} \mid f_{n+1}, Z_{n+1}; \mathcal{S}^n) \tag{3}$$

- **Property (3)**: With respect to the ordering of atoms in $\mathcal{S}^n$, the map $p^{\text{focus}}$ should be permutation-equivariant while $p^{\text{species}}$ and $p^{\text{position}}$ should be permutation-invariant.

We represent $p^{\text{focus}}, p^{\text{species}}$ and $p^{\text{position}}$ as probability distributions because there may be multiple valid choices of focus $f_{n+1}$, species $Z_{n+1}$ and position $\vec{\mathbf{r}}_{n+1}$.

### 3.3 THE DESIGN OF SYMPHONY

The overall working of Symphony is shown graphically in Figure 1. Symphony first computes atom embeddings via an EMBEDDER. Here, we assume that $\text{EMBEDDER}(\mathcal{S}^n) = \{h_{v,l} \mid v \in \mathcal{S}^n, 0 \leq l \leq l_{\max}\}$ returns a set of $E(3)$-equivariant features $h_v, l$ of degree $l$ upto a predefined degree $l_{\max}$, for each atom $v$ in $\mathcal{S}^n$. In Theorem B.1, we show that Symphony can guarantee **Properties (1), (2)** and **(3)** as long as EMBEDDER is permutation-equivariant and $E(3)$-equivariant.

From **Property (1)**, $p^{\text{focus}}$ and $p^{\text{species}}$ should be invariant under rotation and translations of $\mathcal{S}^n$. Since the atom types and the atom indices are discrete sets, we can represent both of these distributions as a vector of probabilities. Thus, we compute $p^{\text{focus}}$ and $p^{\text{species}}$ by applying a multi-layer perceptron MLP on only the rotation and translation invariant features of $\text{EMBEDDER}(\mathcal{S}^n)$:

$$p^{\text{focus}}(f_{n+1}; \mathcal{S}^n) = \text{MLP}(\text{EMBEDDER}(\mathcal{S}^n)_{f_{n+1}, 0}) \tag{4}$$

$$p^{\text{species}}(Z_{n+1} \mid f_{n+1}; \mathcal{S}^n) = \text{MLP}(\text{EMBEDATOMTYPE}(Z_{n+1}) \cdot \text{EMBEDDER}(\mathcal{S}^n)_{f_{n+1}, 0}) \tag{5}$$

Alongside the node-wise probabilities for $p^{\text{focus}}$, we also predict a global STOP probability, indicating that no atom should be added.

On the other hand, **Property (2)** shows that $p^{\text{position}}$ transforms non-identically under rotations and translations. We describe a novel parametrization of 3D probability densities such as $p^{\text{position}}$ with spherical harmonic projections.

The position $\vec{\mathbf{r}}$ is represented by spherical coordinates $(r, \theta, \phi)$ where $r$ is the distance from the focus $f$, $\theta$ is the polar angle and $\phi$ is the azimuthal angle. Any probability distribution $p^{\text{position}}$ over positions must satisfy the normalization and non-negativity constraints: $\int_\Omega p^{\text{position}}(r, \theta, \phi) \, dV = 1$ and $p^{\text{position}}(r, \theta, \phi) \geq 0$ where $dV = r dr \sin\theta d\theta d\phi$ is the volume element and $\Omega = \{r \in [0, \infty), \theta \in [0, \pi], \phi \in [0, 2\pi)\}$ represents all space in spherical coordinates. Since these constraints are hard to incorporate directly into a neural network, we predict the unnormalized logits $f^{\text{position}}(r, \theta, \phi)$ instead, and take the softmax over all space: $p^{\text{position}}(r, \theta, \phi) = \frac{1}{Z} \exp f^{\text{position}}(r, \theta, \phi)$ To model these logits, we first discretize the radial component $r$ into a set of discrete values. We choose 64 uniformly spaced values from 0.9A to 2.0A, which covers all of the bond lengths in QM9. For each fixed value of $r$, we obtain a function on the sphere $S^2$, which we represent in the basis of spherical harmonic functions $Y_{l,m}(\theta, \phi)$, as described in Section 2 and similar to the construction of Cohen & Welling (2015). As we have a radial component $r$ here, the coefficients $c_l$ also depend on $r$:

$$f^{\text{position}}(r, \theta, \phi \mid f_{n+1}, Z_{n+1}; \mathcal{S}^n) = \sum_{l=0}^\infty c_l(r; f_{n+1}, Z_{n+1}, \mathcal{S}^n)^T Y_l(\theta, \phi)$$

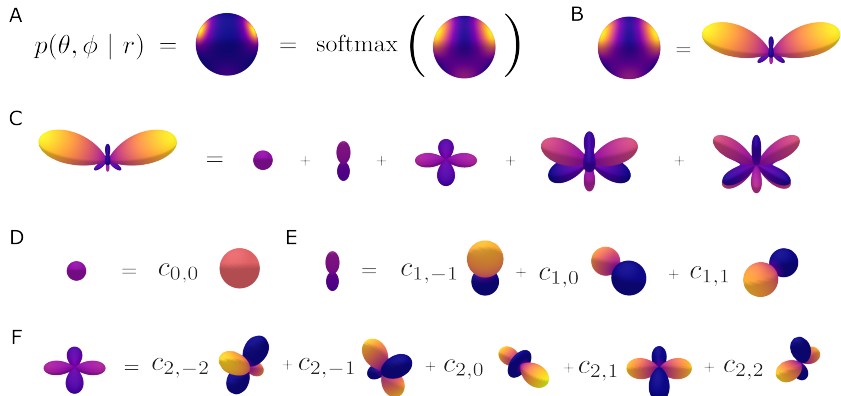

Figure 3: Symphony represents the atom position logits using radial shells of spherical signals. (A) illustrates an example angular distribution for a given radial shell prior to applying the softmax function. The softmax enhances signal peaks and prediction precision. (B) presents two ways to plot the signals for a single shell: as a colored sphere, or as a surface where distance from the origin represents signal magnitude, enhancing peak visibility. (C) breaks the pre-activated signal into contributions from $l = 0$ to $l = 4$ spherical harmonics. (D), (E), and (F) further break these contributions into the $2l + 1$ spherical harmonics for $l = 0, 1$ and $2$.

Symphony predicts these coefficients $c_l$ from the degree $l$ features of the focus node $\text{EMBEDDER}(\mathcal{S}^n)_{f_{n+1},l}$, and the embedding of the target species $Z_{n+1}$:

$$c_l(r; f_{n+1}, Z_{n+1}, \mathcal{S}^n) = \text{LINEAR}(\text{EMBEDDER}(\mathcal{S}^n)_{f_{n+1},l} \otimes \text{EMBEDATOMTYPE}(Z_{n+1})) \quad (6)$$

By explicitly modelling the probability distributions $p^{\text{focus}}$, $p^{\text{species}}$ and $p^{\text{position}}$, Symphony learns to represent all possible options of completing $\mathcal{S}^n$ into a valid molecule.

## 3.4 Bypassing the Angular Frequency Bottleneck

For computational reasons, we are often limited to using a finite number of spherical harmonic projections (ie, up to some $l_{\max}$). Due to the way the spherical harmonics are constructed, this means we can only represent signals upto some angular frequency. For example, to represent a signal on the sphere with peaks separated by $d$ radians, we need spherical harmonic projections with $l_{\max} \geq \frac{2\pi}{d}$. This is similar to issues faced when using the first few terms of the Fourier series; we cannot represent high frequency components. To bypass the bottleneck of angular frequency, we propose using *multiple* channels of spherical harmonic projections, which are then summed over after a non-linearity: $f^{\text{position}}(r, \theta, \phi; \mathcal{S}^n) = \log \sum_{\text{channel ch}} \exp \sum_{l=0}^{\infty} c_l^{\text{ch}}(r; \mathcal{S}^n)^T Y_l(\theta, \phi)$. See Appendix F for a concrete example where adding multiple channels effectively increases the angular frequency capacity of our model. For Symphony, we find that $2$ channels is sufficient, as demonstrated in

## 3.5 Training and Inference

We utilize teacher forcing to train Symphony. At training time, the true focus $f_{n+1}$ and atomic number $Z_{n+1}$ are provided as computed in NEXTFRAGMENT. Thus, no sampling occurs at training time. The true probability distributions $q^{\text{focus}}$ and $q^{\text{species}}$ are computed empirically from the set of unfinished atoms and their corresponding neighbors in $\mathcal{M}$. The true probability distribution $q^{\text{position}}$ is computed by smoothly approximating a Dirac delta distribution upto some cutoff frequency $l_{\max}$ at the target position $\vec{r}_{n+1}$ around the focus atom. Further details about the training process and representing Dirac delta distributions are provided in Section C.2 and Appendix H.

$$q^{\text{position}}(\vec{r}) = \frac{1}{Z} \exp\left(-\frac{\|\vec{r}\| - \|\vec{r}_{n+1}\|}{2\sigma_{\text{true}}^2} \cdot \delta_{l_{\max}}\left(\hat{\mathbf{r}} - \hat{\mathbf{r}}_{n+1}\right)\right) \quad (7)$$

At inference time, both the focus $f_{n+1}$ and atomic number $Z_{n+1}$ are sampled from $p^{\text{focus}}(\cdot; \mathcal{S}^n)$ and $p^{\text{species}}(\cdot | f_{n+1}; \mathcal{S}^n)$ respectively. These are used to sample $\vec{r}_{n+1}$ from $p^{\text{position}}(\cdot | f_{n+1}, Z_{n+1}; \mathcal{S}^n)$.

Molecules are generated by starting from an initial fragment $\mathcal{S}^1$, and repeatedly sampling from $p^{\text{focus}}$, $p^{\text{species}}$ and $p^{\text{position}}$ until a STOP is predicted or $N_{\max} = 35$ iterations have occurred, based on the maximum size of molecules in the QM9 dataset as 30 atoms.

### 3.6 RELATION TO PRIOR WORK

Most methods for 3D molecular structure generation fall into one of two broad categories: autoregressive and end-to-end models. G-SchNet (Gebauer et al., 2019; 2022) and G-SphereNet (Luo & Ji, 2022) were the first successful attempts at autoregressive generation of molecular structures.

G-SchNet uses the SchNet framework (Schütt et al., 2017) to perform message-passing with rotationally invariant features and compute node embeddings. A focus node is then selected as the center of a 3D grid. All of the atoms in the current fragment then vote on where to place the next atom within this grid by specifying a radial distance to the next atom. Because of the use of only rotationally invariant features, at least three atoms are needed to be present in the current fragment to specify the exact position of the next atom without any degeneracy due to symmetry; this procedure is called *triangulation*. This requires several additional tokens to break symmetry. Similarly, G-SphereNet learns a normalizing flow to perform a triangulation procedure once there are atleast 3 atoms in $\mathcal{S}^n$.

We wish to highlight two observations that guided the development of Symphony:

- Rotationally invariant features centered at a single point cannot capture the orientations of geometrical motifs (Pozdnyakov & Ceriotti, 2022). To handle the degeneracies inherent when using rotationally invariant features to predict positions, G-SchNet uses unphysical auxiliary tokens (which are multiple spatial positions that are not atoms) to break symmetry.

- G-SchNet queries all of the atoms in $\mathcal{S}^n$ at each iteration, which means distant atoms can have an undue influence when placing the next atom. Similarly, G- SphereNet predictions are not a smooth function of the input fragment; when the input is perturbed slightly, the choice of atoms used in the triangulation procedure can change drastically.

Recently, $E(3)$-equivariant neural networks that build higher-degree $E(3)$-equivariant features have demonstrated improved performance on a wide range of atomistic tasks (Batzner et al., 2022; Geiger & Smidt, 2022; Owen et al., 2023). Our key contribution is to show the benefit of higher-degree $E(3)$-equivariant features for the *molecular generation* task allowing for a novel parametrization of 3D probability distributions using spherical harmonic projections. Simm et al. (2021) also uses spherical harmonic projections with a single channel for molecule generation, but trained with reinforcement learning, and sampled using rejection sampling. We discuss these details in Appendix D.

Among end-to-end generation methods, Hoogeboom et al. (2022) developed EDM, a state-of-the-art $E(3)$-equivariant diffusion model. EDM significantly outperformed the previously proposed $E(3)$-equivariant normalizing flow (ENF) models for molecule generation (Satorras et al., 2022a). EDM learns to gradually denoise a initial configuration of atoms into a valid molecular structure. Both EDM and ENF are built on the $E(n)$-Equivariant Graph Neural Networks (Satorras et al., 2022b) framework which can utilize only scalar and vector features (and interactions between them). MiDi (Vignac et al., 2023) improves EDM by utilizing bond order information (and hence, a 2D molecular graph to compare to), which we do not assume access to here. While expressive, diffusion models are expensive to train, requiring $\approx 3.5\times$ more training on the QM9 dataset to outperform autoregressive models. Unlike autoregressive models, diffusion models do not flexibly allow for completion of molecular fragments, because they are usually trained in setups where all atoms are free to move. Current diffusion models use fully-connected graphs where all atoms interact with each other. This could potentially affect their scalability when building larger molecules. On the other hand, Symphony and other autoregressive models use distance cutoffs to restrict interactions and improve efficiency. In Appendix I, we find that EDM with distance cutoffs performs quite poorly.

Furthermore, diffusion models are significantly slower to sample from, because the underlying neural network is invoked $\approx 1000$ times when sampling a single molecule. Flow matching (Lipman et al., 2023) has also emerged as a competitor for diffusion models for 3D molecule generation (Song et al., 2023), but suffers from the same drawbacks listed above.

## 4 EXPERIMENTAL RESULTS

A major challenge with generative modelling is evaluating the quality of generated 3D structures. Ideally, a generative model should generate physically plausible structures, accurately capture training set statistics and generalize well to molecules outside of its training set. We propose a comprehensive set of tests to evaluate Symphony and other generative models along these three aspects.

### 4.1 VALIDITY OF GENERATED STRUCTURES

All of the generative models considered here output a set of atoms with 3D coordinates; bonding information is not generated by the model. Before we can use cheminformatics tools designed for molecules, we need to assign bonds between atoms. Multiple algorithms exist for bond order assignment: `xyz2mol` (Kim & Kim, 2015), OpenBabel (Banck et al., 2011) and a simple lookup table based on empirical pairwise distances in organic compounds (Hoogeboom et al., 2022). Here, we perform the first comparison between these algorithms for evaluating machine-learning generated 3D structures. In Table 1, we use each of these algorithms to infer the bonds and create a molecule from generated 3D molecular structure. We declare a molecule as valid if the algorithm could successfully assign bond order with no net resulting charge. We also measure the uniqueness to see how many repetitions were present in the set of SMILES (Weininger, 1988) strings of valid generated molecules. Ideally, we want both the validity and the uniqueness to be high.

While EDM (Hoogeboom et al., 2022) is still superior on the validity and uniqueness metrics, we find that Symphony performs much better on both validity and uniqueness than existing autoregressive models, G-SchNet (Gebauer et al., 2019) and G-SphereNet (Luo & Ji, 2022), for the `xyz2mol` and OpenBabel algorithms. Note that the lookup table does not account for aromatic bonds and is quite sensitive to exact bond lengths; we believe this penalizes Symphony due to its coarser discretization compared to EDM and G-SchNet. Of note is that only `xyz2mol` finds almost all of the ground truth QM9 structures to be valid.

| Metric ↑ | QM9 | Symphony | EDM | G-SchNet | G-SphereNet |
|---|---|---|---|---|---|
| Validity via `xyz2mol` | 99.99 | 83.50 | **86.74** | 74.97 | 26.92 |
| Validity via OpenBabel | 94.60 | 74.69 | **77.75** | 61.83 | 9.86 |
| Validity via Lookup Table | 97.60 | 68.11 | **90.77** | 80.13 | 16.36 |
| Uniqueness via `xyz2mol` | 99.84 | 97.98 | **99.16** | 96.73 | 21.69 |
| Uniqueness via OpenBabel | 99.97 | 99.61 | **99.95** | 98.71 | 7.51 |
| Uniqueness via Lookup Table | 99.89 | 97.68 | **98.64** | 93.20 | 23.29 |

Table 1: Validity and uniqueness (among valid) percentages of molecules.

Recently, Buttenschoen et al. (2023) showed that the predicted 3D structures from machine-learned protein-ligand docking models tend to be highly unphysical. For Table 2, we utilize their PoseBusters framework to perform the following sanity checks to count how many of the predicted 3D structures are reasonable. We see that the valid molecules from all models tend to be quite reasonable, with Symphony performing better than all baselines on generating structures with reasonable UFF (Rappe et al., 1992) energies and respecting the geometry constraints of double bonds. Further details about the PoseBusters tests are provided in Section E.1.

| Test ↑ | Symphony | EDM | G-SchNet | G-SphereNet |
|---|---|---|---|---|
| All Atoms Connected | 99.92 | 99.88 | 99.87 | **100.00** |
| Reasonable Bond Angles | 99.56 | **99.98** | 99.88 | 97.59 |
| Reasonable Bond Lengths | 98.72 | **100.00** | 99.93 | 72.99 |
| Aromatic Ring Flatness | **100.00** | **100.00** | 99.95 | 99.85 |
| Double Bond Flatness | **99.07** | 98.58 | 97.96 | 95.99 |
| Reasonable Internal Energy | **95.65** | 94.88 | 95.04 | 36.07 |
| No Internal Steric Clash | 98.16 | **99.79** | 99.57 | 98.07 |

Table 2: Percentage of valid (as obtained from `xyz2mol`) molecules passing each PoseBusters test.

### 4.2 Capturing Training Set Statistics

Next, we evaluate models on how well they capture bonding patterns and the geometry of local environments found in the training set molecules. In previous work (Luo & Ji, 2022; Hoogeboom et al., 2022), models were compared based on how well they capture the true bond length distributions observed in QM9. However, such statistics only deal with pairwise bond lengths and cannot capture the geometry of how atoms are placed relative to each other. Here, we utilize the *bispectrum* (Uhrin, 2021) as a rotationally invariant descriptor of the geometry of local environments. Given a local environment with a central atom $u$, we first project all of the neighbors of $u$ according to the inferred bonds onto the unit sphere $S^2$. Then, we compute the signal $f$ as a sum of Dirac delta distributions along the direction of each neighbor: $f(\hat{\mathbf{r}}) = \sum_{v \in N(u)} \delta_{l_{\max}} (\hat{\mathbf{r}} - \hat{\mathbf{r}}_{vu})$. The bispectrum $\mathcal{B}(f)$ of $f$ is then defined as: $\mathcal{B}(f) = \text{ExtractScalars}(f \otimes f \otimes f)$. Thus, $f$ captures the distribution of atoms around $u$, and the bispectrum $\mathcal{B}(f)$ captures the geometry of this distribution. The advantage of the bispectrum is that it varies smoothly when $f$ is varied and is guaranteed to be rotationally invariant. We compute the bispectrum of local environments with atleast 2 neighboring atoms. Note that we exclude the pseudoscalars in the bispectra.

For comparing discrete distributions, we use the symmetric Jensen-Shannon divergence (JSD) as employed in Hoogeboom et al. (2022). Given the true distribution $Q$ and the predicted distribution $P$, the Jensen-Shannon divergence between them is defined as: $D_{JS}(Q \,\|\, P) = \frac{1}{2} D_{KL}(Q \,\|\, M) + \frac{1}{2} D_{KL}(P \,\|\, M)$ where $D_{KL}$ is the Kullback–Leibler divergence and $M = \frac{Q+P}{2}$ is the mean distribution. For continuous distributions, estimating the Jensen-Shannon divergence from samples is tricky without further assumptions on the distributions. Instead, we use the Maximum Mean Discrepancy (MMD) score from Luo & Ji (2022) instead to compare samples from continuous distributions. The MMD score is the distance between means of features computed from samples from the true distribution $Q$ and the predicted distribution $P$. A model with a smaller MMD score captures the true distribution of samples better. We provide details about the MMD score in Section E.2.

From Table 3 we see that Symphony and other autoregressive models struggle to match the bond length distribution of QM9 as well as EDM. This is the case except for the single C-H and single N-H bonds. On the bispectra, however, Symphony attains the lowest MMD for several environments. To gain some intuition for these MMD numbers, we also plotted the bond length distributions, samples of the bispectra, atom type distributions and other statistics in Appendix A for each model.

### 4.3 Generalization Capabilities

All of the metrics discussed so far can be maximized by simply memorizing the training set molecules. Now, we propose a new metric to evaluate how well the models have actually learned to generate valid chemical structures. We compare models by asking them to complete fragments of 1000 unseen molecules from the test set, with one hydrogen atom removed. We then check how many final molecules were deemed valid. Since the valid completion rate (VCR) depends heavily on the quality of the model, we compute the valid completion rate for fragments of molecules from the training set as well. If the performance is significantly different between the two sets of fragments, this indicates that the models do not generalize well. Diffusion models such as EDM are more challenging to evaluate for this task, since we would need a way to fix the initial set of atoms, so we compare only Symphony and G-SchNet. Encouragingly, both models are able to generalize well to unseen fragments, but Symphony's overall completion rate is higher for both seen and unseen fragments. We notice that the performance of Symphony on this task seems to decrease as training progresses, the reason for which remains unclear.

### 4.4 Molecule Generation Throughput

One of the major advantages of autoregressive models (such as Symphony) over diffusion models (such as EDM) is significantly faster inference speeds. As measured on a single NVIDIA RTX A5000 GPU, Symphony's inference speed is 0.293 seconds/molecule, compared to EDM's 0.930 sec/mol. Symphony is much slower than existing autoregressive models (G-SchNet is at 0.011 sec/mol, and G-SphereNet 0.006) because of the additional tensor products for generating higher-degree $E(3)$-equivariant features, but is still approximately $3\times$ faster than EDM. However, our sampler is currently bottlenecked by some of the limitations of JAX (Bradbury et al., 2018); we believe that Symphony's inference speed reported here can be significantly improved to match its training speed.

| MMD of Bond Lengths ↓ | Symphony | EDM | G-SchNet | G-SphereNet |
|---|---|---|---|---|
| C-H: 1.0 | 0.0739 | **0.0653** | 0.3817 | 0.1334 |
| C-C: 1.0 | 0.3254 | **0.0956** | 0.2530 | 1.0503 |
| C-O: 1.0 | 0.2571 | **0.0757** | 0.5315 | 0.6082 |
| C-N: 1.0 | 0.3086 | **0.1755** | 0.2999 | 0.4279 |
| N-H: 1.0 | **0.1032** | 0.1137 | 0.5968 | 0.1660 |
| C-O: 2.0 | 0.3033 | **0.0668** | 0.2628 | 2.0812 |
| C-N: 1.5 | 0.3707 | **0.1736** | 0.5828 | 0.4949 |
| O-H: 1.0 | 0.2872 | 0.1545 | 0.7899 | **0.1307** |
| C-C: 1.5 | 0.4142 | **0.1749** | 0.2051 | 0.8574 |
| C-N: 2.0 | 0.5938 | **0.3237** | 0.4194 | 2.1197 |

| MMD of Bispectra ↓ | Symphony | EDM | G-SchNet | G-SphereNet |
|---|---|---|---|---|
| C: C2,H2 | 0.2165 | **0.1003** | 0.4333 | 0.6210 |
| C: C1,H3 | 0.2668 | **0.0025** | 0.0640 | 1.2004 |
| C: C3,H1 | **0.1111** | 0.2254 | 0.2045 | 1.1209 |
| C: C2,H1,O1 | **0.1500** | 0.2059 | 0.1732 | 0.8361 |
| C: C1,H2,O1 | 0.3300 | 0.1082 | **0.0954** | 1.6772 |
| O: C1,H1 | 0.0282 | 0.0056 | 0.0487 | **0.0030** |
| C: C2,H1,N1 | **0.1481** | 0.1521 | 0.1967 | 1.3461 |
| C: C2,H1 | 0.2525 | **0.0468** | 0.1788 | 0.2403 |
| C: C1,H2,N1 | 0.3631 | 0.2728 | **0.1610** | 0.9171 |
| N: C2,H1 | **0.0953** | 0.2339 | 0.2105 | 0.6141 |

| Jensen-Shannon Divergence ↓ | Symphony | EDM | G-SchNet | G-SphereNet |
|---|---|---|---|---|
| Atom Type Counts | 0.0003 | **0.0002** | 0.0011 | 0.0026 |
| Local Environment Counts | **0.0039** | 0.0057 | 0.0150 | 0.1016 |

Table 3: Comparing statistics of generated molecules to those found in QM9. (Top): The MMD of bond lengths for the 10 most frequent bonds. The notation 'X-Y: T' means that a X atom was bonded to a Y atom with a bond of type T. (Middle): The MMD of bispectra for the 10 most occurring local environments. The notation 'X: $Yn,Zm$' means that an X atom was the central atom, surrounded by $n$ Y atoms and $m$ Z atoms. (Bottom): The JSD of occurrence counts for atom types and local environments. ↓ indicates that lower is better for the metrics.

| Valid Completion Rate ↑ | Symphony 500K steps | Symphony 800K steps | Symphony 1000K steps | G-SchNet |
|---|---|---|---|---|
| Training: $VCR_{train}$ | **98.53** | 96.65 | 95.57 | 97.91 |
| Testing: $VCR_{test}$ | **98.66** | 96.30 | 95.43 | 98.15 |

Table 4: Comparing the difference between fragment completion rates on (seen) training and (unseen) testing fragments with one hydrogen removed.

## 5 CONCLUSION

We have proposed Symphony, a new method to autoregressively generate 3D molecular geometries with spherical harmonic projections and higher-degree $E(3)$-equivariant features. We show promising results on molecular generation and completion, relative to existing autoregressive models. However, one drawback of our current formulation is that the discretization of our radial components is too coarse, so our bond length distributions are not as accurate as EDM or G-SchNet. This affects our validity when using lookup tables to assign bond orders as they are particularly sensitive to exact bond lengths. Further, Symphony incurs increased computational cost due to the use of tensor products to create higher degree $E(3)$-equivariant features. As a highlight, Symphony is trained on only $\approx 80$ epochs, while G-SchNet and EDM are trained for 330 and 1100 epochs respectively. Further exploring the data efficiency of Symphony remains to be seen. In the future, we plan to explore normalizing flows to smoothly model the radial distribution without any discretization, and placing entire local environment motifs at once which would speed up generation.

# 6 REPRODUCIBILITY STATEMENT

Our JAX code containing all of the data preprocessing, model training and evaluation metrics is available at `https://github.com/atomicarchitects/symphony`.

Section C.2 describes the hyperparameters used in the training process for Symphony. Details about the metrics used can be found in Appendix E. Section C.3 contains all of the information regarding the QM9 dataset used in this work. Further information about the baseline models and the sampled structures can be found in Section C.4.

# 7 ETHICS STATEMENT

Generative models for molecules such as Symphony have the potential to be used for discovering novel drugs and useful catalysts. While harmful uses of such generative models exist, the synthesis of a molecule given only its 3D geometry is still extremely challenging. Thus, we do not anticipate any negative consequences of our research.

ACKNOWLEDGMENTS

Ameya Daigavane was supported by the National Science Foundation under Cooperative Agreement PHY-2019786 (The NSF AI Institute for Artificial Intelligence and Fundamental Interactions), and the NSF Graduate Research Fellowship program. Song Kim was supported by Analog Devices as an Undergraduate Research and Innovation Scholar in the MIT Advanced Undergraduate Research Opportunities Program (SuperUROP). Mario Geiger and Tess Smidt were supported by the Integrated Computational and Data Infrastructure (ICDI) program of the U.S. Department of Energy, grant number DE-SC0022215. The authors acknowledge the MIT SuperCloud and Lincoln Laboratory Supercomputing Center for providing HPC resources that have contributed to the research results reported within this paper.

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

APPENDIX

## A    ADDITIONAL ANALYSES

For all of the analyses performed in this section, we used all the valid molecules for each model as computed by `xyz2mol`.

### A.1    BISPECTRA OF LOCAL ENVIRONMENTS IN SAMPLED MOLECULES

As seen in Figure 4, we see that Symphony's sampled bispectra (second from left) have a slightly different distribution relative to those from QM9 in the two most frequent local environments.

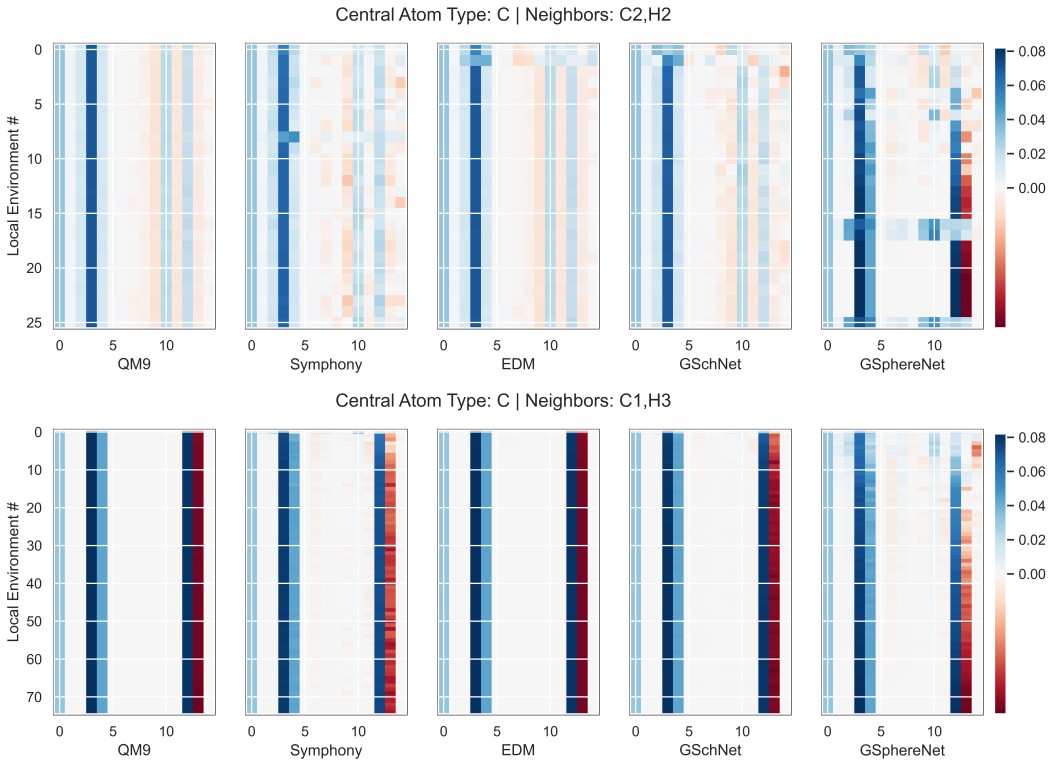

Figure 4: Bispectra of local environments of type C: C2,H2 and type C: C1,H3 respectively. Each row corresponds to a sample of the bispectrum (an array of length 15). Every entry of the bispectra is colored by value according to the colorbar on the right.

### A.2    BOND LENGTHS IN SAMPLED MOLECULES

From Figure 5 and Figure 6, we see that Symphony's bond length distribution tends to be wider than those of QM9, hurting its MMD score relative to EDM. Improving this aspect is an ongoing effort; but we believe that the bond lengths are still quite reasonable.

### A.3    ATOM TYPE COUNTS

As seen in Figure 7, all models are able to reasonably capture the distribution of atom types in QM9; Symphony performs especially well here.

### A.4    RING SIZES

We also extracted all rings using RDKit (Landrum et al., 2023) and counted their relative frequency, in Figure 8. G-SphereNet seems to produce either very large or very small rings. The other models seem to capture the distribution of ring sizes well.

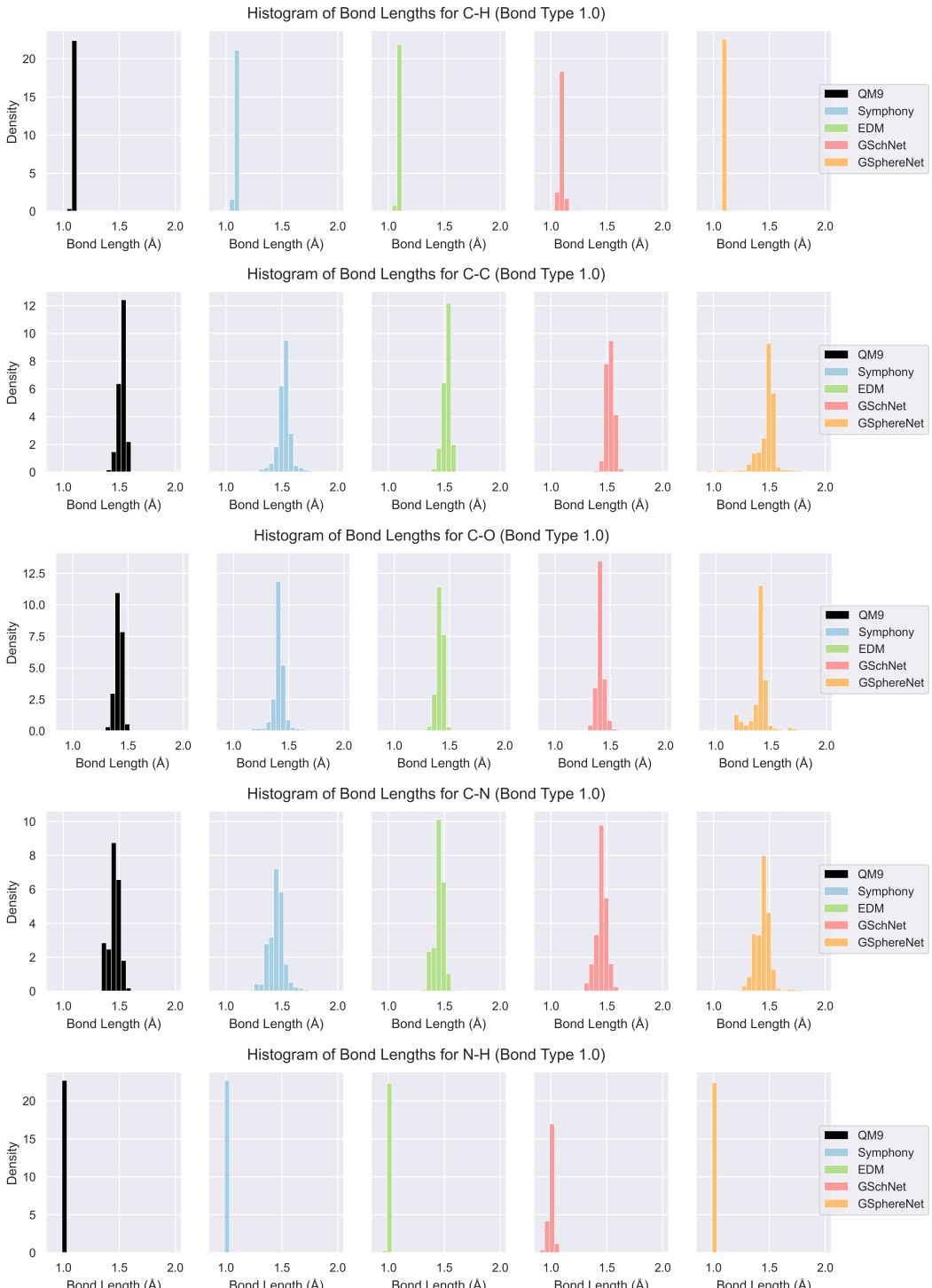

Figure 5: Histogram of bond lengths for the five most frequent bonds in QM9.

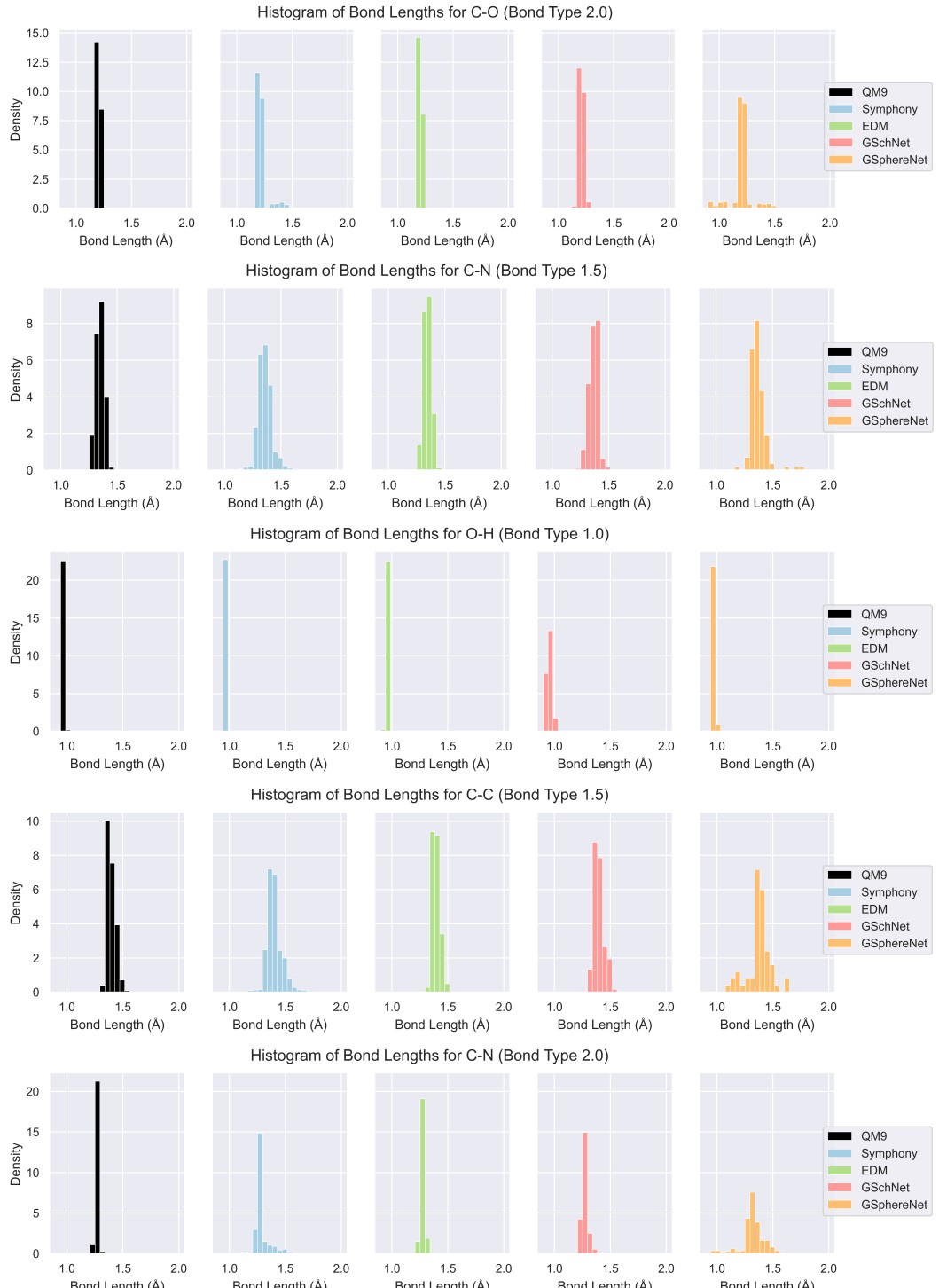

Figure 6: Histogram of bond lengths for the sixth to tenth most frequent bonds in QM9.

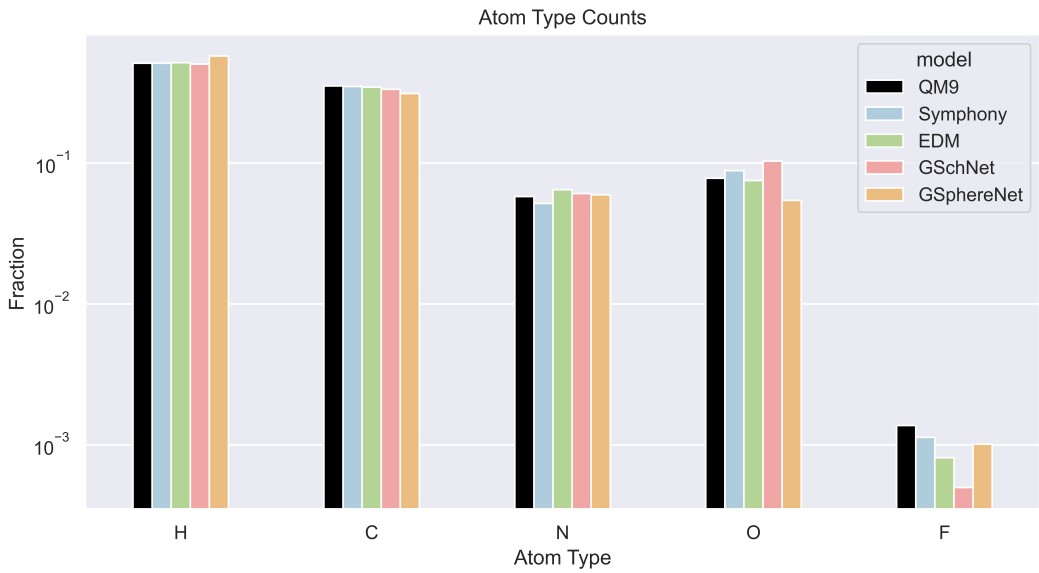

Figure 7: Frequency of atom type counts in generated molecules on a log-scale.

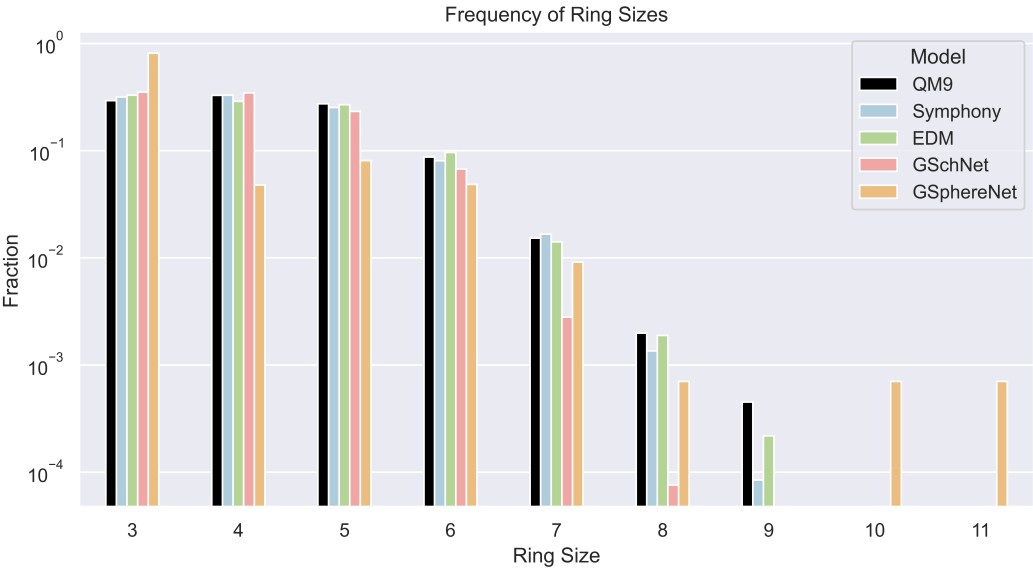

Figure 8: Frequency of ring sizes in generated molecules on a log-scale.

## B    PROOF OF E(3)-EQUIVARIANCE

**Theorem B.1.** Suppose EMBEDDER produces $O(3)$-equivariant and translation-invariant features $h_{v,l} = \text{EMBEDDER}(\mathcal{S}^n)_{v,l}$ for every atom $v$. Then, $p^{\text{position}}$ is $O(3)$-equivariant and translation-invariant (and hence, $E(3)$-equivariant):

$$p^{\text{position}}(\mathbf{R}\vec{\mathbf{r}}_{n+1} + \mathbf{T} \mid f_{n+1}, Z_{n+1}; \mathbf{R}\mathcal{S}^n + \mathbf{T}) = p^{\text{position}}(\vec{\mathbf{r}}_{n+1} \mid f_{n+1}, Z_{n+1}; \mathcal{S}^n)$$

**Proof**: We first show that $p^{\text{position}}$ is $O(3)$-equivariant. We have:

$$\text{EMBEDDER}(\mathbf{R}\mathcal{S}^n)_{v,l} = D^l(\mathbf{R})^T \text{EMBEDDER}(\mathcal{S}^n)_{v,l}$$

for every atom $v$ and degree $l$. Note that because $Z_{n+1}$ is rotationally invariant, it immediately follows from Equation 6 and the above, that $c_l$ is also $E(3)$-equivariant with degree $l$:

$$c_l(r; \mathbf{R}\mathcal{S}^n, f_{n+1}, Z_{n+1}) = c_l(r; \mathcal{S}^n, f_{n+1}, Z_{n+1})$$

Now, as the Wigner D-matrices are always unitary, we have:

$$
\begin{aligned}
f^{\text{position}}(\mathbf{R}\vec{\mathbf{r}}; \mathbf{R}\mathcal{S}^n, f_{n+1}, Z_{n+1}) &= \sum_{l=0}^{\infty} c_l(r; \mathbf{R}\mathcal{S}^n, f_{n+1}, Z_{n+1})^T Y_l(\mathbf{R}\hat{\mathbf{r}}_{ij}) \\
&= \sum_{l=0}^{\infty} c_l(r; \mathcal{S}^n, f_{n+1}, Z_{n+1})^T D^l(\mathbf{R})D^l(\mathbf{R})^T Y_l(\hat{\mathbf{r}}_{ij}) \\
&= \sum_{l=0}^{\infty} c_l(r; \mathcal{S}^n, f_{n+1}, Z_{n+1})^T Y_l(\hat{\mathbf{r}}_{ij}) \\
&= f^{\text{position}}(\vec{\mathbf{r}}; \mathcal{S}^n)
\end{aligned}
$$

by definition. Thus, we are guaranteed that $f^{\text{position}}$ is $O(3)$-equivariant. Note that applying a pointwise non-linearity ($\exp$) to $f^{\text{position}}$ and a rotationally invariant normalization does not change $O(3)$-equivariance. Thus, $p^{\text{position}}$ is $O(3)$-equivariant as well.

For translations, note that $p^{\text{position}}$ is described relative to the focus atom $f_{n+1}$. Thus, as EMBEDDER is translation-invariant:

$$\text{EMBEDDER}(\mathcal{S}^n + \mathbf{T})_{v,l} = \text{EMBEDDER}(\mathcal{S}^n)_{v,l}$$

$p^{\text{position}}$ will be translation-equivariant:

$$p^{\text{position}}(\vec{\mathbf{r}}_{n+1} + \mathbf{T} \mid f_{n+1}, Z_{n+1}; \mathcal{S}^n + \mathbf{T}) = p^{\text{position}}(\vec{\mathbf{r}}_{n+1} \mid f_{n+1}, Z_{n+1}; \mathcal{S}^n)$$

In conclusion, $p^{\text{position}}$ is $O(3)$-equivariant and translation-equivariant, and hence $E(3)$-equivariant. Thus, **Property (2)** is satisfied. ∎

**Theorem B.2.** Suppose EMBEDDER produces permutation-equivariant features $h_{v,l} = \text{EMBEDDER}(\mathcal{S}^n)_{v,l}$ for every atom $v$. Then, $p^{\text{focus}}$ is permutation-equivariant, while $p^{\text{species}}$ and $p^{\text{position}}$ are permutation-invariant:

$$p^{\text{focus}}(\pi(f_{n+1}); \pi\mathcal{S}^n) = p^{\text{focus}}(f_{n+1}; \mathcal{S}^n)$$
$$p^{\text{species}}(Z_{n+1} \mid \pi(f_{n+1}); \pi\mathcal{S}^n) = p^{\text{species}}(Z_{n+1} \mid f_{n+1}; \mathcal{S}^n)$$
$$p^{\text{position}}(\vec{\mathbf{r}}_{n+1} \mid \pi(f_{n+1}), Z_{n+1}; \pi\mathcal{S}^n) = p^{\text{position}}(\vec{\mathbf{r}}_{n+1} \mid f_{n+1}, Z_{n+1}; \mathcal{S}^n)$$

where $\pi$ represents a permutation of the atoms of $\mathcal{S}^n$.

**Proof**: Because EMBEDDER is permutation-equivariant:

$$\text{EMBEDDER}(\pi\mathcal{S}^n)_{\pi(v),l} = \text{EMBEDDER}(\mathcal{S}^n)_{v,l}$$

for each atom $v$. Then, from Equation 5:

$$
\begin{aligned}
p^{\text{focus}}(\pi(f_{n+1}); \pi\mathcal{S}^n) &= \text{MLP}(\text{EMBEDDER}(\pi\mathcal{S}^n)_{\pi(f_{n+1}),0}) \\
&= \text{MLP}(\text{EMBEDDER}(\mathcal{S}^n)_{f_{n+1},0}) \\
&= p^{\text{focus}}(f_{n+1}); \mathcal{S}^n)
\end{aligned}
$$

as claimed. Similarly,

$$
\begin{aligned}
p^{\text{species}}(Z_{n+1} \mid \pi(f_{n+1}); \pi \mathcal{S}^n) &= \text{MLP}(\text{EMBEDATOMTYPE}(Z_{n+1}) \cdot \text{EMBEDDER}(\pi \mathcal{S}^n)_{\pi(f_{n+1}),0}) \\
&= \text{MLP}(\text{EMBEDATOMTYPE}(Z_{n+1}) \cdot \text{EMBEDDER}(\mathcal{S}^n)_{f_{n+1},0}) \\
&= p^{\text{species}}(Z_{n+1} \mid f_{n+1}; \mathcal{S}^n)
\end{aligned}
$$

For $p^{\text{position}}$, it is sufficient to show that the coefficients $c_l(r)$ are permutation-equivariant:

$$
\begin{aligned}
c_l(r; \pi(f_{n+1}), Z_{n+1}, \pi \mathcal{S}^n) &= \text{LINEAR}(\text{EMBEDDER}(\pi \mathcal{S}^n)_{\pi(f_{n+1}),l} \otimes \text{EMBEDATOMTYPE}(Z_{n+1})) \\
&= \text{LINEAR}(\text{EMBEDDER}(\mathcal{S}^n)_{f_{n+1},l} \otimes \text{EMBEDATOMTYPE}(Z_{n+1})) \\
&= c_l(r; f_{n+1}, Z_{n+1}, \mathcal{S}^n)
\end{aligned}
$$

Thus, all distributions transform as expected. ∎

## C  DETAILS OF MODELS

### C.1  EMBEDDERS

Here, we describe E3SchNet and NequIP (Batzner et al., 2022) which we use to embed the atoms in each fragment into $E(3)$-equivariant features. As shown in Appendix B, we require these models to be $E(3)$-equivariant.

Both of these models are geometric message-passing neural networks, a type of graph neural network (Sanchez-Lengeling et al., 2021; Daigavane et al., 2021) that respects the symmetries of 3D structures. In particular, E3SchNet as the EMBEDDER for the focus and atom type prediction, and NequIP as the EMBEDDER for the position prediction. Unlike previous autoregressive models which utilized a shared embedder for all tasks, we found that using different embedders for these two tasks performed much better in our experiments.

Given the fragment $\mathcal{S}^n$, we define the neighbour of each atom $i \in \mathcal{S}^n$ by a Euclidean distance cutoff $\leq d_{\max}$:

$$
\mathcal{N}(i) = \{j \in \mathcal{S}^n \mid \|\vec{\mathbf{r}}_{ij}\| \leq d_{\max}\} \tag{8}
$$

Initially, the features $h_i^{(0)}$ of each atom $i$ in $\mathcal{S}^n$ are set as the embedding of its atomic number $Z_i$. At each iteration $t$, the features $h_i^{(t)}$ is updated using the atom's features $h_i^{(t-1)}$ and its neighbour's features $h_j^{(t-1)}$ where $j \in \mathcal{N}(i)$ from the previous round. The final embedding for atom $i$ is returned as $h_i^{(T)}$ where $T$ is the number of message-passing iterations. Algorithm 2 formally shows the operations of a general message passing neural network.

---

**Algorithm 2** General Operation of a Message Passing Neural Network

---

**Input:** Fragment $\mathcal{S}^n$, Message Passing Iterations $T$, Cutoff $d_{\max}$, Update Function UPDATE
    Compute neighbor lists for each atom in $\mathcal{S}^n$ according to Equation 8.
    **for** $i = 1, 2, \ldots, n$ **do**:
        $h_i^{(0)} \leftarrow \text{SCALAREMBEDDING}(Z_i)$
    **for** $t = 1, 2, \ldots, T$ **do**:
        **for** $i = 1, 2, \ldots, n$ **do**:
            $h_{\mathcal{N}(i)}^{(t-1)} \leftarrow \{h_j^{(t-1)} \mid j \in \mathcal{N}(i)\}$
            $h_i^{(t)} \leftarrow \text{UPDATE}(h_i^{(t-1)}, h_{\mathcal{N}(i)}^{(t-1)})$
    **return** $\{h_i^{(T)}\}_{i=1}^n$

---

Different message-passing networks differ in their choice of UPDATE function. Following Batzner et al. (2022), the UPDATE for NequIP is defined as:

$$
\text{UPDATE}(h_i^{(t-1)}, h_{\mathcal{N}(i)}^{(t-1)}) = h_i^{(t-1)} + \frac{1}{C} \sum_{j \in \mathcal{N}(i)} \sum_{l=0}^{l_{\max}} R_\Theta(\|\vec{\mathbf{r}}_{ij}\|) Y^l(\hat{\mathbf{r}}_{ij}) \otimes h_j^{(t-1)}
$$

$R_\Theta(\cdot)$ is a learned multi-layer perceptron (MLP). We set $C = 20, d_{\max} = 5\text{A}, l_{\max} = 5$, and $T = 3$ here. For clarity, we assume the decomposition of the tensor product into a direct sum of irreducible representations of $O(3)$ above.

E3SchNet is our generalization of the SchNet model (Schütt et al., 2017) that was used in (Gebauer et al., 2019) to produce higher-degree $E(3)$-equivariant features. The UPDATE function for E3SchNet is defined as:

$$\text{UPDATE}(h_i^{(t-1)}, h_{\mathcal{N}(i)}^{(t-1)}) = h_i^{(t-1)} + \text{LINEAR}\left(\sum_{j \in \mathcal{N}(i)} \sum_{l=0}^{l_{\max}} W_{ijl} \cdot \left(h_j^{(t-1)} \otimes Y^l(\hat{\mathbf{r}}_{ij})\right)\right)$$

where $W_{ijl}$ are scalars computed via:

$$W_{ijl} = \text{LINEAR}(\sigma(\text{CUTOFF}(\|\vec{\mathbf{r}}_{ij}\|) \cdot \text{RADIALBASIS}(\|\vec{\mathbf{r}}_{ij}\|)))$$

We use the Gaussian radial basis functions, following SchNet. In fact, for $l_{\max} = 0$, E3SchNet reduces exactly to the standard SchNet. We set $l_{\max} = 2$, as we find that the benefits of using even higher degree features for the focus and atom type prediction task are minimal. The cutoff is again 5A.

We see that NequIP and E3SchNet guarantee permutation-equivariance, translation invariance and $O(3)$-equivariance, and hence satisfy the requirements for EMBEDDER in Appendix B.

We implement Symphony with the `e3nn-jax` library that utilizes the JAX (Bradbury et al., 2018) framework for creating efficient $E(3)$-equivariant machine learning models.

## C.2 TRAINING DETAILS

We set $\sigma_{\text{true}}^2 = 10^{-5}$ and express the Dirac delta distribution in the spherical harmonic basis upto $l_{\max} = 5$, as explained in Appendix H. The predicted distributions $p^{\text{focus}}, p^{\text{species}}$ and $p^{\text{position}}$ are learned by minimizing the KL divergence to their true counterparts. We found that adding a small amount of zero-centered Gaussian noise $\sigma^2 = 2.5 \times 10^{-3}$ to all input atom positions helped with robustness. All parameters in the EMBEDDER, MLP and LINEAR layers are trained with the Adam (Kingma & Ba, 2017) optimizer with a learning rate of $5 \times 10^{-4}$. We chose the parameters that achieved the lowest loss on the validation set over 8000000 training steps with a batch size of 16 fragments.

## C.3 DATA DETAILS

Following EDM (Hoogeboom et al., 2022), we obtained the QM9 (Rupp et al., 2012) dataset using the DeepChem library (Ramsundar et al., 2019), and filtered out 3054 'uncharacterized' molecules (available at `https://springernature.figshare.com/ndownloader/files/3195404`) which rearranged significantly during geometry optimization, giving us exactly 130831 molecules. Symphony was trained used the same splits as EDM: 100000 molecules to train, 13083 molecules for validation and 17748 molecules for test, obtained from a random permutation of the molecules.

## C.4 BASELINE MODEL DETAILS

For the baseline models, we used the pretrained EDM model at `https://github.com/ehoogeboom/e3_diffusion_for_molecules` and the pretrained G-SphereNet model at `https://github.com/divelab/DIG/tree/dig-stable/examples/ggraph3D/G_SphereNet`. We retrained the G-SchNet model on the EDM splits following `https://github.com/atomistic-machine-learning/G-SchNet`. The samples (in `.xyz` format) of all models used for evaluation is available at this URL: `https://figshare.com/s/a17ccface17f0c22f15a`.

## D    LEARNING AND SAMPLING FROM POSITION DISTRIBUTIONS

In this section, we drop the superscript from $p^{\text{position}}$ as it should be clear from context.

## D.1 TRAINING

To recap Section 3.3, Symphony predicts coefficients $c_l^{\text{ch}}(r; \mathcal{S}^n)$ to represent the position distribution $p$:

$$f(r, \theta, \phi) = \log \sum_{\text{channel ch}} \exp \sum_{l=0}^{\infty} c_l^{\text{ch}}(r; \mathcal{S}^n)^T Y_l(\theta, \phi)$$

$$p(r, \theta, \phi) = \frac{1}{Z} \exp f(r, \theta, \phi)$$

where $Z$ is the partition function.

As mentioned in Section C.2, the coefficients are learned by minimizing the KL divergence to the target distribution $q$:

$$KL(q \parallel p) = \int_\Omega q(\vec{r}) \log \frac{q(\vec{r})}{p(\vec{r})} d\vec{r} = \int_\Omega q(\vec{r}) \log q(\vec{r}) d\vec{r} - \int_\Omega q(\vec{r}) f(\vec{r}) d\vec{r} + \log Z$$

Following the notation of Section 3.3, $\Omega$ represents the set $\{r \in [0, \infty), \theta \in [0, \pi], \phi \in [0, 2\pi)\}$ which is all space in spherical coordinates.

For training, we only need the unnormalized logits $f$ and not the normalized distribution $p$. This is identical to the log-sum-exp trick when training with cross-entropy loss for a classification problem. Unlike the classification case where the number of classes is finite, the integral above must be computed over all of $r$, $\theta$ and $\phi$ which is an infinite set. To numerically approximate this integral, we use a uniform grid on $r$ and a Spherical Gauss-Legendre quadrature on the sphere at each value of $r$. As discussed in Section 3.3, the uniform grid on $r$ spans $64$ values from $0.9$A to $2.0$A which is more than sufficient to cover all bond lengths in organic molecules. The Spherical Gauss-Legendre quadrature is a product of two quadratures: a 1D Gauss-Legendre quadrature with $180$ points over $\cos\theta \in [-1, 1]$, and a uniform grid of $359$ points over $[0, 2\pi)$ for $\phi$.

Symphony predicts the coefficients $c_l(r)$ of $f$ which can be used to evaluate $f(r, \theta, \phi)$ at any point. This evaluation for a spherical grid of $(\theta, \phi)$ values can be done quickly via a Fast Fourier Transform (FFT) that is implemented in `e3nn-jax`. We perform this FFT procedure for each sphere defined by a radial grid point $r$.

## D.2 SAMPLING

Once the model is learnt, we need to sample from the distribution $p$. A key advantage of predicting the coefficients $c_l(r)$ of $f_\theta(r, \theta, \phi)$ is that a different resolution of angular grid can be chosen for sampling than that of training. We simply evaluate $f(r, \theta, \phi)$ on the quadrature grid as before, apply the exponential, and normalize via numerical integration to get $p(r, \theta, \phi)$. We first marginalize over $\theta, \phi$ to obtain a distribution $p(r)$ to sample a radius $r$. Then, we sample one of the angular grid points $(\theta, \phi)$ for the sphere corresponding to this radius $r$. Overall, this procedure gives us a sample from $p(r, \theta, \phi)$.

In Section G.2, we assess how the validity of molecules generated by Symphony varies as the grid resolution is varied.

Note that our sampling procedure is much simpler than that of Simm et al. (2021), which uses rejection sampling with a uniform base distribution. We perform some quantitative experiments with the parametrization of Simm et al. (2021) in Section F.2.

While we are primarily interested in learning distributions over $\mathbb{R}^3$ which are equivariant under $E(3)$, there has been prior work in learning distributions over manifolds Cohen & Welling (2015); Murphy et al. (2022), where the issue of estimating the partition function are also solved by discretizing over an appropriate domain.

# E DETAILS OF METRICS

## E.1 POSEBUSTERS

Table 5 provides details of the Posebusters tests used in Table 2. We use the default parameters from their framework.

| Test | Description |
|------|-------------|
| All Atoms Connected | There exists a path along bonds between any two atoms in the molecule. |
| Reasonable Bond Lengths | The bond lengths in the input molecule are within $0.75$ of the lower and $1.25$ of the upper bounds determined by distance geometry. |
| Reasonable Bond Angles | The angles in the input molecule are within $0.75$ of the lower and $1.25$ of the upper bounds determined by distance geometry. |
| Aromatic Rings Flatness | All atoms in aromatic rings with 5 or 6 members are within $0.25$A of the closest shared plane. |
| Double Bonds Flatness | The two carbons of aliphatic carbon-carbon double bonds and their four neighbours are within $0.25$A of the closest shared plane. |
| Reasonable Molecule Energy | The calculated energy of the input molecule is no more than $100$ times the average energy of an ensemble of $50$ conformations generated for the input molecule. The energy is calculated using the UFF (Rappe et al., 1992) in RDKit (Landrum et al., 2023) and the conformations are generated with ETKDGv3 (Riniker & Landrum, 2015) followed by force field relaxation using the UFF with up to 200 iterations. |
| No Internal Steric Clash | The interatomic distance between pairs of non-covalently bound atoms is above $0.8$ of the lower bound determined by distance geometry. |

Table 5: Description of each intramolecular PoseBusters test, taken from Table 4 of Buttenschoen et al. (2023).

### E.2 MAXIMUM MEAN DISCREPANCY

The Maximum Mean Discrepancy (MMD), introduced in Gretton et al. (2012), measures how different two distributions $p_X$ and $p_Y$ are, given a kernel function $k$. Formally, the MMD is defined as:

$$\text{MMD}(p_X, p_Y) = \sqrt{\mathop{\mathbb{E}}_{X,X'\sim p_X}[k(X,X')] + \mathop{\mathbb{E}}_{Y,Y'\sim p_X}[k(Y,Y')] - \mathop{\mathbb{E}}_{X\sim p_X, Y\sim p_Y}[k(X,Y)]}$$

From the above equation, we see that the MMD can be easily estimated with samples from each distribution. We choose $k$ as the sum of Gaussian kernels at different scales:

$$k(X, X') = \sum_{i=0}^{29} \exp(-10^{\left(\frac{i}{5}-3\right)} \cdot \|X - X'\|^2)$$

## F THE ADVANTAGE OF USING MULTIPLE CHANNELS OF SPHERICAL HARMONICS

### F.1 AN EXAMPLE WITH THE OCTAHEDRON

Figure 9 shows how adding a second channel helps reduce the effective $l_{\max}$ needed to represent $p^{\text{position}}$. The atoms depicted by red circles have been placed already, and the atom at the center of the octahedron has been chosen as the focus. To accurately capture the positions of the three remaining atoms (depicted by two stars and a square), we would need a projection upto $l_{\max} = 4$, because the angle made by the 'star', central atom and the 'square' is $\frac{\pi}{2}$ radians. However, if we used one channel to represent the 'stars' and one to represent the 'square', we can get away by only using projections upto $l_{\max} = 2$, because the 'stars' are diametrically opposite each other.

### F.2 A STUDY ON LEARNING RANDOM SIGNALS

To quantitatively show the effect of having multiple channels, we see how well the model is able to learn a random distribution on the sphere. We randomly sample $N = 5$ target points with coordinates

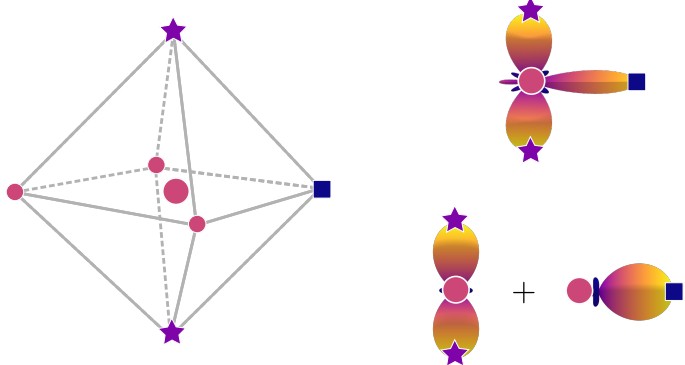

Figure 9: Usually, we would require $l_{\max} = 4$ to represent $p^{\text{position}}$ for the 'stars' and 'square' atoms, centered at the red central atom. With two channels, we only need up to $l_{\max} = 2$ each.

$\{\hat{\mathbf{r}}_i\}_{i=1}^{N}$ on the sphere, and then define the distribution:

$$q(\hat{\mathbf{r}}) = \sum_{i=1}^{N} \exp(\delta_{l_{\max}}(\hat{\mathbf{r}} - \hat{\mathbf{r}}_i))$$

with the same Dirac delta distribution approximation as described in Appendix H. We use $l_{\max} = 5$ throughout this section. Then, we randomly initialize coefficients $c$ to minimize the KL divergence to $q$:

$$\min_{c} KL(q \,||\, p_c)$$

where $p_c$ is the probability distribution defined by coefficients $c$, as before:

$$f(\theta, \phi) = \log \sum_{\text{channel ch}} \exp \sum_{l=0}^{l_{\max}} c_l^{\text{ch}\,T} Y_l(\theta, \phi)$$

$$p(\theta, \phi) = \frac{1}{Z} \exp f(\theta, \phi)$$

This corresponds to a simpler setting where we have only one radius $r$.

We assess the KL divergence as a function of number of position channels ch and $l_{\max}$ in Figure 10. We see a consistent improvement across different $l_{\max}$ as the number of position channels are increased.

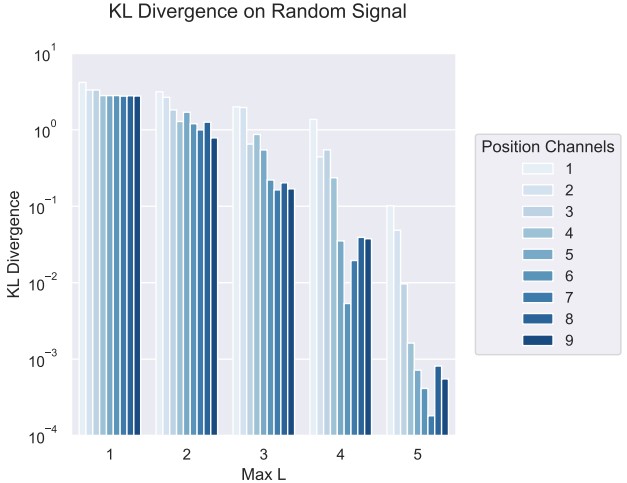

Figure 10: Final KL divergence $KL(q \,||\, p_c)$ for learned coefficients $c$ as a function of number of position channels ch and $l_{\max}$.

We also experimented with the parametrization from Simm et al. (2021), who define:

$$p(\theta, \phi) = \frac{1}{Z} \exp\left(-\frac{\beta}{k}|f(\theta, \phi)|^2\right)$$

where $k = \sum_{l=0}^{l_{max}} |c_l|^2$. This extra factor of $k$ was proposed by Simm et al. (2021) to "regularize the distribution so that it does not approach a delta function". In the left panel of Figure 11, we show that this regularization hurts the model. Even adding multiple channels does not help, because the regularization term 'switches' off multiples channels. However, as shown in the right panel of Figure 11, removing this regularization significantly helps the model, with further improvement as the number of channels are increased. For $l_{max} = 5$, we see that our parametrization performs similarly to Simm et al. (2021) without the regularization term. Based on this experiment, we plan to experiment with non-linearities for the logits in future versions of Symphony.

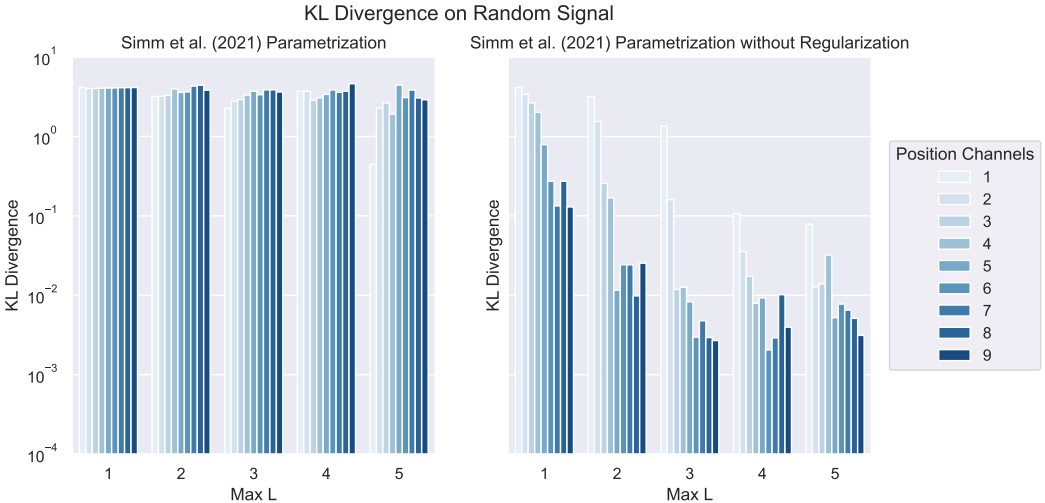

Figure 11: Final KL divergence $KL(q \,||\, p_c)$ for learned coefficients $c$ as a function of number of position channels ch and $l_{max}$, with the parametrization proposed by Simm et al. (2021). Removing the regularization term helps the model learn better.

# G  ABLATION STUDIES

## G.1  $l_{MAX}$ AND NUMBER OF POSITION CHANNELS

To understand the practical effect of adding multiple position channels to Symphony, as well as the impact of increasing $l_{max}$, we trained variants of Symphony varying $l_{max}$ for the focus embedder E3SchNet from 1 to 2, the number of position channels from 1 to 4, and $l_{max}$ for the position embedder NequIP from 1 to 5.

Due to computational constraints, we trained these models for $1,000,000$ steps each, which is $8\times$ lesser than the original model reported in Section 4. Thus, the validity numbers are slightly lower overall. However, we believe we can still observe important trends from this experiment.

We report the validity as measured by `xyz2mol` for each of these models in Figure 12.

- For the focus embedder E3SchNet, we do not see a significant increase in validity when going from $l_{max} = 1$ to $l_{max} = 2$.

- For the position embedder NequIP, we find a large jump when going from $l_{max} = 1$ to $l_{max} = 2$. Further increasing $l_{max}$ seemed to help slightly. For computational reasons, we kept $l_{max} = 5$.

- Increasing the number of position channels helps for $l_{max} = 1$ in particular.

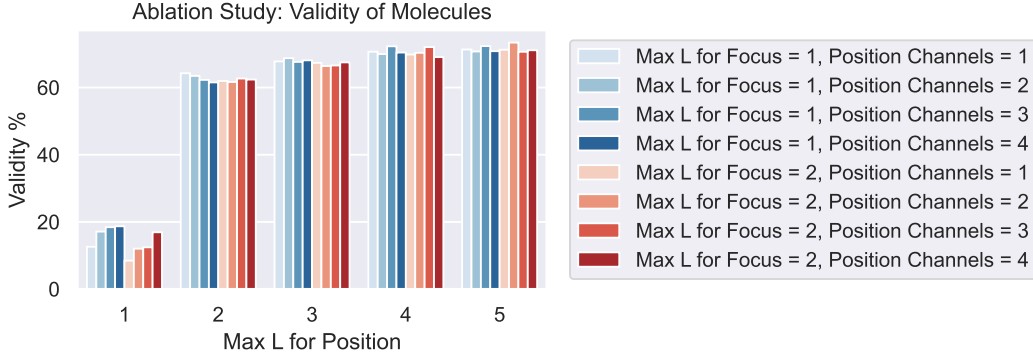

Figure 12: Validity as a function of $l_{\max}$ for the position and focus embedders. Models for which $l_{\max} = 1$ for the focus embedder are marked in blue. Models for which $l_{\max} = 2$ for the focus embedder are marked in red. The intensity of colours increases with the number of position channels.

### G.2 RESOLUTION

Here, we take the trained Symphony model, freeze all weights, and measure the validity of molecules across a range of grid resolutions. The original grid resolution for model training was $(r_\theta, r_\phi) = (180, 359)$ as described above. From Figure 13, we see that the validity is within the expected variation even when using upto $10\times$ smaller grids. Further amplification of the resolution also does not seem to affect the validity. We hypothesize that this is due to sampling with a lower temperature than ideal making the target distribution more diffuse; future work will seek to understand this effect better.

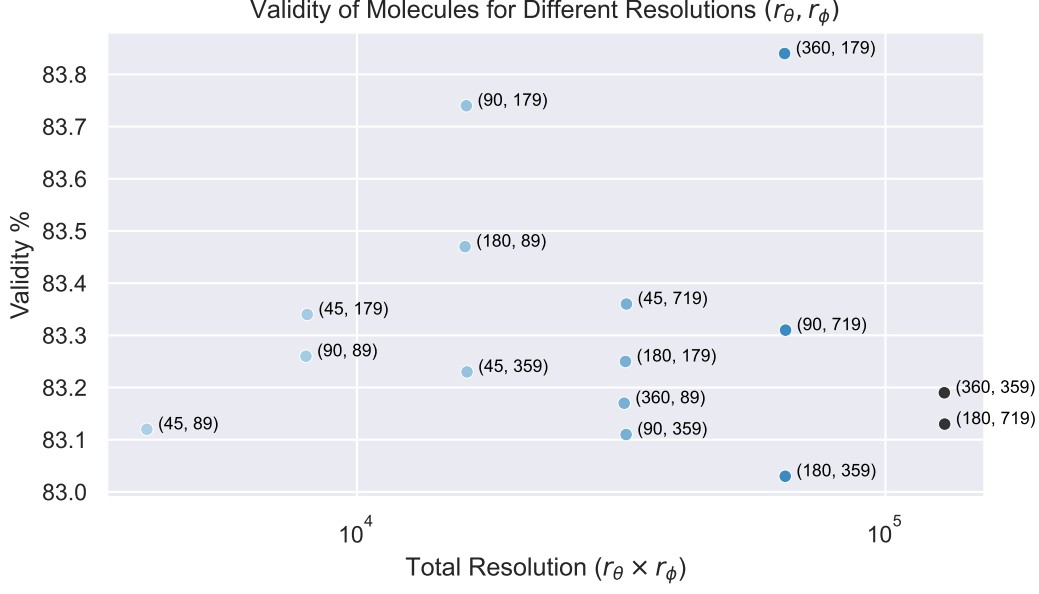

Figure 13: Validity as a function of sampling grid resolution $(r_\theta, r_\phi)$.

The previous experiment measured the effect of the grid resolution for sampling. We also sought to understand the effect of the grid resolution for training. For this, we reuse the task of Section F.2, and vary the grid resolution. All other hyperparameters were kept fixed, with $l_{\max} = 2$ and 2 position channels. From Figure 14, we see that the learning is not affected even at low resolutions. In fact, from a KL divergence perspective, it is easier to learn at lower resolutions because localization is easier. However, lower resolutions come with decreased accuracy when sampling, as shown by the rightmost plot of Figure 14.

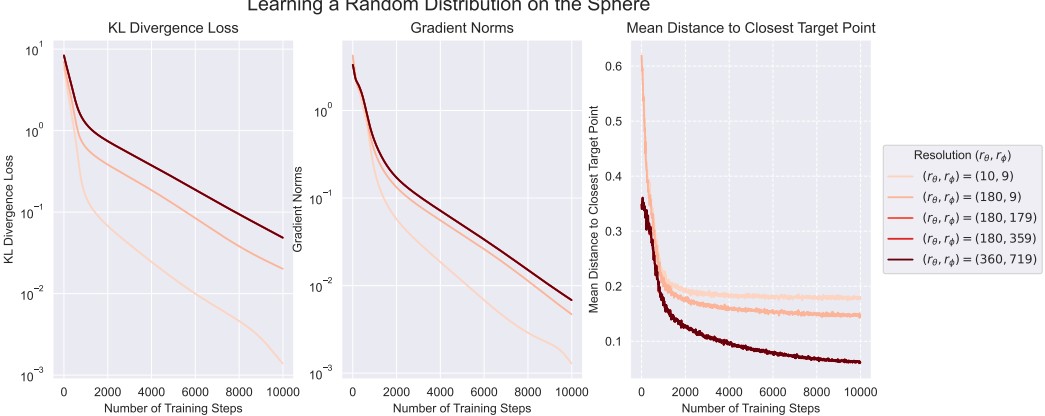

Figure 14: The effect of resolution when learning the random signal from Section F.2. Our original model was trained with a resolution of $(r_\theta, r_\phi) = (180, 359)$.

### G.3 TEMPERATURE

Again, we take the trained Symphony model, freeze all weights, and measure the validity of molecules across a range of temperatures $T$. This means scaling all the logits by a factor of $\frac{1}{T}$. Higher temperatures make the model more diffuse, while lower temperatures make the model more peaked. We see that while the validity improves significantly at lower temperatures, the uniqueness tends to suffer. As seen in Figure 15, this experiment suggests a more careful sampling of the temperature to better understand a Pareto frontier between validity and uniqueness.

## H REPRESENTING DIRAC DELTA DISTRIBUTIONS

Suppose we have the function $f(\hat{\mathbf{r}}) = \delta(\hat{\mathbf{r}} - \hat{\mathbf{r}}_0)$ defined on the sphere $S^2$, and we wish to compute its spherical harmonic coefficients $c_{l,m}$:

$$f(\theta, \phi) = \sum_{l=0}^{l_{\max}} c_l^T Y_l(\theta, \phi) = \sum_{l=0}^{l_{\max}} \sum_{m=-l}^{l} c_{l,m} Y_{l,m}(\theta, \phi)$$

By orthonormality of the spherical harmonics, and the annihilation property of the Dirac delta:

$$c_{l,m} = \int f(\theta, \phi) Y_{l,m}(\theta, \phi) \sin\theta d\theta d\phi$$

$$= \int \delta(\hat{\mathbf{r}} - \hat{\mathbf{r}}_0) Y_{l,m}(\theta, \phi) \sin\theta d\theta d\phi$$

$$= Y_{l,m}(\hat{\mathbf{r}}_0)$$

Thus, we can easily compute the spherical harmonic coefficients for the Dirac delta distribution upto any required $l_{\max}$. This is implemented in the `e3nn-jax` package. Due to the frequency cutoff, the Dirac delta distribution thus obtained is a smooth approximation of a true Dirac delta.

## I THE EFFECT OF DISTANCE CUTOFFS ON DIFFUSION MODELS

As described in Section 3.6, EDM (Hoogeboom et al., 2022) and other diffusion models use fully-connected graphs to denoise the 3D molecular graph at each timestep. Here, we investigate the performance of EDM as we apply a distance-based cutoff $c$ to the edges:

$$(i, j) \in E \iff ||\vec{\mathbf{r}}_i - \vec{\mathbf{r}}_j|| \leq c.$$

In Table 6, we vary $c$ on the QM9 dataset to measure how many of the edges in the fully-connected graphs are kept after the radial cutoff.

In Figure 16, we measured the 'atom stability' as computed by Hoogeboom et al. (2022) on the EDM-generated molecules as $c$ is varied, when trained on QM9. We notice a sharp drop-off in this

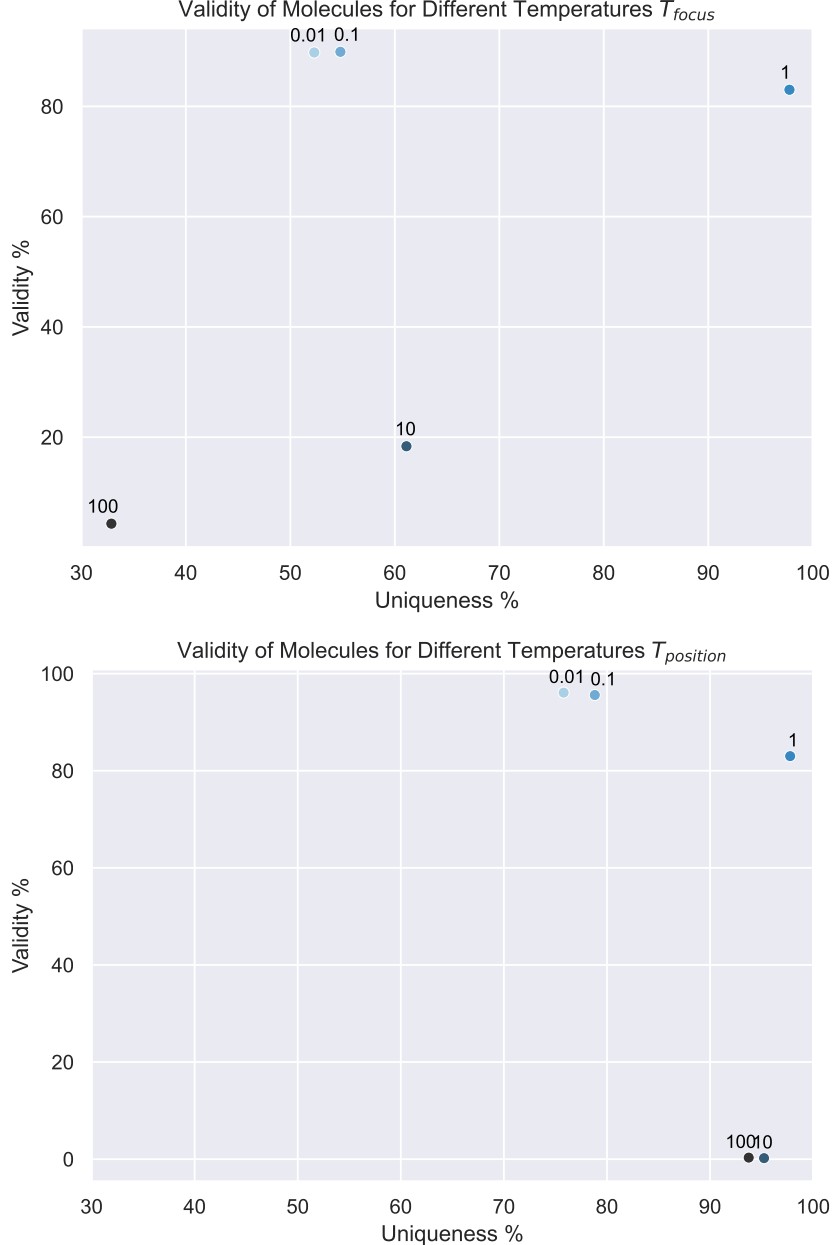

Figure 15: Validity as a function of temperature applied to the focus (above) and position (below) distribution logits.

| Radial Distance Cutoff | Percentage of Edges Preserved |
|---|---|
| 2.0 A | $\approx 26\%$ |
| 3.0 A | $\approx 61\%$ |
| 5.0 A | $\approx 93\%$ |
| 10.0 A | $100\%$ |

Table 6: Percentage of edges preserved by the radial distance cutoff for molecular graphs from QM9.

metric as $c$ (and hence, the number of edges) reduces, despite the use of many message-passing layers in the underlying graph neural network. Even for $c = 5$ A, where $93\%$ of the edges in the graph still remain, the quality of the EDM-generated molecules drops significantly. On observing the trajectories of the individual atoms, we noticed that the system tends to explode for lower values of $c$. Note that Symphony uses a radial cutoff of $5$ A as well.

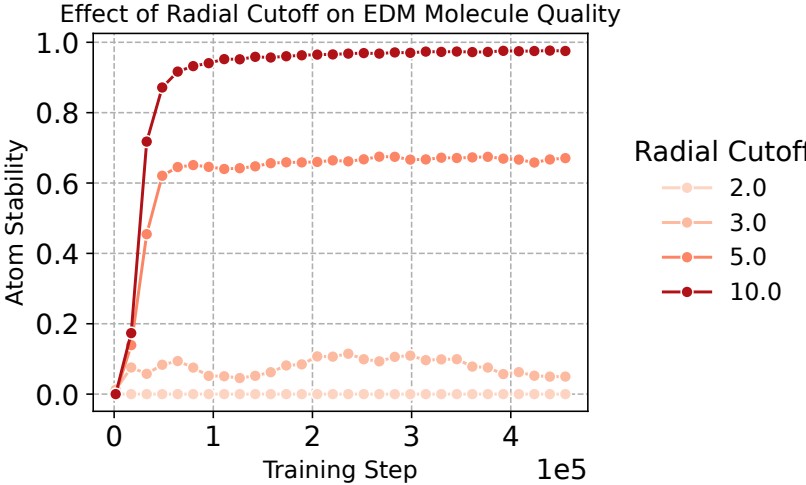

Figure 16: Atom stability on QM9 as measured by Hoogeboom et al. (2022) as a function of radial cutoff $c$, showing that the model is quite sensitive.

## J    GENERATED MOLECULES FROM SYMPHONY

Figure 17 exhibits random non-cherry-picked samples from Symphony.

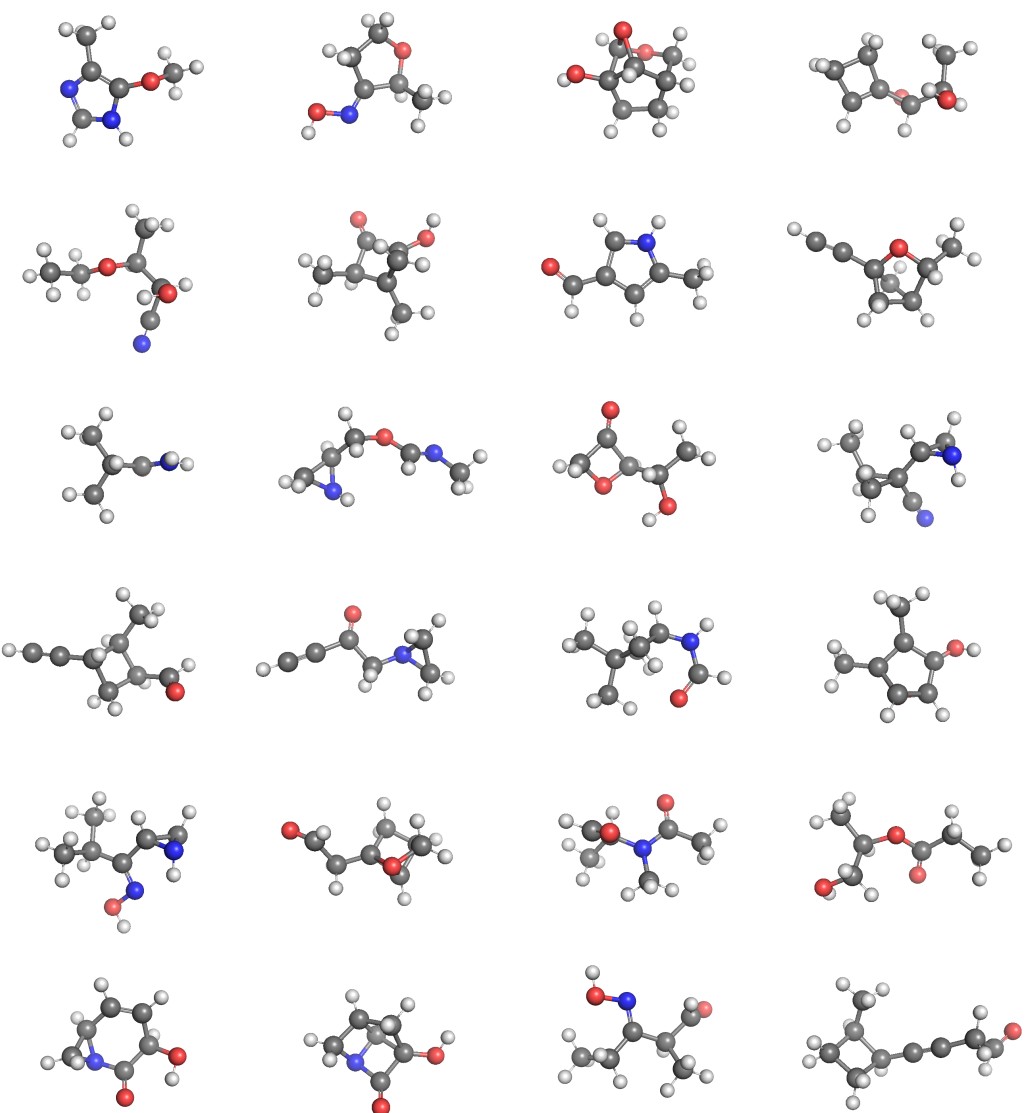

Figure 17: Molecules generated by Symphony and visualized with PyMOL (Schrödinger, LLC, 2015).

