# OpenReview forum: "Symphony: Symmetry-Equivariant Point-Centered Spherical Harmonics for 3D Molecule Generation"
_ICLR.cc/2024/Conference — ICLR 2024 poster_

### Official Review · Reviewer_msmy · 2023-10-29

**Soundness:** 3 good
**Presentation:** 3 good
**Contribution:** 3 good
**Rating:** 6
**Confidence:** 3

**Summary:**

This manuscript introduces Symphony, an autoregressive technique designed for the generation of 3D molecular structures via spherical harmonic projections. Unlike prevailing autoregressive models like G-SchNet and G-SphereNet, which employ rotationally invariant features of degree 1, Symphony leverages higher-degree E(3) equivariant features. The proposed model demonstrates superior performance in generating small molecules from the QM9 dataset and offers competitive results when compared to the E(3)-equivariant diffusion model.

**Strengths:**

1. This paper is well written and well organized.
2. Distinct from other autoregressive equivariant generative models, the proposed approach employs higher-degree E(3) equivariant features. This enhances the flexibility in representing probability distributions.
3. In terms of performance, the proposed method is on par with existing diffusion models. However, it boasts a significant advantage in computational efficiency, as current diffusion models rely on fully-connected graphs that may pose scalability challenges.

**Weaknesses:**

1. The manuscript employs a coarse discretization of the radial component for predicting the relative positions of subsequent atoms, a limitation acknowledged in the conclusion section. It would be intriguing to explore the effects of employing a finer discretization. Specifically, would such a refinement significantly compromise the computational efficiency of the method? Furthermore, could this potentially improve the distribution of bond lengths?
2. n Equation (1), would it be more precise to describe this as a conditional distribution of $f$ given $S$? does the embedder at $f_{n+1}$ use both $h^{\text{position}}$ and $h^{\text{focus}}$?
3. Typo: In the final equation on page 4, the term should correctly be denoted as $p^{\text{position}}$.
4. On the first line of page 6, the mechanism for predicting the STOP condition remains unspecified. Could you elaborate on this aspect?
5. Typo: The first equation in Section 2 misses a translation $T$.

**Questions:**

Please refer to the previous section

---

> ### Author Response · Authors · 2023-11-18
>
> Thank you for your positive review! We answer specific questions below. Please also take a look at our common response.
>
> > would such a refinement significantly compromise the computational efficiency of the method?
>
> We answer exactly this question below! The following is the approximate training speed of the model for a batch size of $16$ graphs, as measured on a single NVIDIA RTX A5000 GPU:
>
> | Number of Radii| $32$ | $64$ | $128$ | $256$ |
> | -------- | -------- | -------- | -------- | -------- |
> |  Approximate Training Speed (steps/second)     |   $16.5$   | $13.8$ | $10.1$ | $6.6$ |
>
> So while there definitely is a tradeoff, it does not seem to be a major bottleneck. We are currently exploring the effect of the radial discretization on the bond length metrics.
>
> > In Equation (1), would it be more precise to describe this as a conditional distribution of $f$ given $S$?
>
> Hmm, it is just a matter of notation, whether to represent it as $p(f | S)$ or $p(f; S)$. In any case, the fragment $S$ is given at training time, but is sampled from the model's predictions at inference time. So, both options are okay.
>
> > does the embedder at $f_{n + 1}$ use both $h^\text{position}$ and $h^\text{focus}$?
>
> For predicting the focus and the target atom type, $h^\text{focus}$ is used.
> For predicting the positions, $h^\text{position}_f$ is used. The two embedders never interact with each other directly.
>
> > Typo: In the final equation on page 4, the term should correctly be denoted as $p^\text{position}$.
>
> We believe this formulation is correct: we are parametrizing the logits $f^\text{position}$ of the distribution $p^\text{position}$. The logits are then exponentiated to get the distribution, as described in Section 3.3.
>
> > On the first line of page 6, the mechanism for predicting the STOP condition remains unspecified. Could you elaborate on this aspect?
>
> Great find! We have added this to Section 3.3 under Equation 5.
>
> > Typo: The first equation in Section 2 misses a translation $T$.
>
> The features we look at here are invariant under translation. We have clarified this in the text below.
>
> Thank you again!

---

> > ### Comment · Reviewer_msmy · 2023-11-22
> >
> > I thank the authors for the response. My rating remains.

---

### Official Review · Reviewer_cp6D · 2023-11-01

**Soundness:** 3 good
**Presentation:** 3 good
**Contribution:** 3 good
**Rating:** 6
**Confidence:** 4

**Summary:**

The authors present Symphony, an E(3)-equivariant autoregressive generative model, that leverages a unique parametrization of 3D probability densities with spherical harmonic projections to make predictions based on features from a single focus atom. This approach captures both radial and angular distributions of potential atomic positions. Additionally, the model incorporates a novel metric based on the bispectrum to assess the angular accuracy of generated local environments. The authors claim that the model demonstrates superior performance on the QM9 dataset compared to previous autoregressive models and is competitive with existing diffusion models across various metrics.

**Strengths:**

1. Symphony uses a novel parametrization of 3D probability densities with spherical harmonic projections, allowing for predictions based on features from a single focus atom (novelty). This method overcomes the limitations of traditional approaches, such as G-SchNet and G-SphereNet, which require at least three atoms to precisely determine the position of the next atom, eliminating uncertainties due to symmetry.

2. The NN for probability distribution over positions satisfies normalization and non-negativity constraints by applying softmax functions.

3. A novel metric based on the bispectrum is introduced to assess the angular accuracy of matching generated local environments to similar environments in training sets.

4. Symphony is able to generate valid molecules with a high success rate, even when conditioned on unseen molecular fragments.

**Weaknesses:**

1. Complexity: Spherical harmonic projections and inverse projections involve complex mathematical operations, potentially increasing the computational complexity of the model. There is no comparison regarding the time spent by the network among different autoregressive generation models, such as G-SchNet and G-SphereNet.

2. Approximation Error: Spherical harmonic projections are an approximation method and may introduce errors, especially when a limited number of spherical harmonics are used.

3. Dependence on Focus Atom: This method relies on the selection of an appropriate focus atom; if the focus atom is poorly chosen, it may affect the accuracy of predictions. (not specific for the paper).

4. Challenges in Handling Symmetry: When dealing with the symmetry of molecules, ensuring that the predicted distributions are symmetrically equivariant(although proved) may increase the complexity of the model.

5. The experiments are comprehensive, but the improvement is marginal because

(1) In Table 1, the EDM performs best;

(2) In Table 3, the MMD of Bond lengths from Symphony is worse than from EDM.

**Questions:**

See weaknesses.

---

> ### Author Response · Authors · 2023-11-18
> **Response to Reviewer cp6D**
>
> Thank you for your thoughtful feedback! We answer specific questions below. Please also take a look at our common response.
>
> > There is no comparison regarding the time spent by the network among different autoregressive generation models, such as G-SchNet and G-SphereNet.
>
> This comparison is present in Section 4.4 of our manuscript. Symphony is slower than G-SchNet and G-SphereNet. Symphony is currently 3x faster than EDM, but we believe the sampling time for Symphony can be improved by avoiding some of JAX's limitations. We measured the inference times ourselves on the same NVIDIA RTX A5000 GPU.
>
> > Approximation Error: Spherical harmonic projections are an approximation method and may introduce errors, especially when a limited number of spherical harmonics are used.
>
> We show the effect of increasing $l_\max$ and the number of position channels for Symphony variants in Section G.1 (Figure 12). We did not feel that much utility could be obtained on going to $l_\max > 5$. We also measure the error due to the sampling grid by measuring the validity of molecules generated in Section G.2 (Figure 13).
>
> > When dealing with the symmetry of molecules, ensuring that the predicted distributions are symmetrically equivariant(although proved) may increase the complexity of the model.
>
> This is true, but we believe it is necessary that a model should respect the underlying symmetries of a problem. Ours is just one approach, however.

---

### Official Review · Reviewer_XBL9 · 2023-11-06

**Soundness:** 3 good
**Presentation:** 2 fair
**Contribution:** 3 good
**Rating:** 6
**Confidence:** 4

**Summary:**

This submission provides an E(3)-equivariant autoregressive generative model for 3D molecule generation. The use of multiple channels of spherical harmonic projections is novel. The results show the performance is comparable to THE diffusion-based model EDM, but slightly worse.

**Strengths:**

1. The whole framework for autoregressive generation is reasonable: running message-passing on the current molecular fragments (current state), selecting the focus node, predicting atom species, and finally predicting the position of the added atom. This framework is similar to many previous autoregressive models.

2. The main contribution is the definition of the distribution for the position of the added atom. The representation for $r$ by spherical coordinates and the use of spherical harmonic functions is novel and reasonable for the target task.

3. Using multiple channels of spherical harmonic projections is novel and the authors provide valid reasons for using it. However, it lacks an ablation study to show how the multiple channels influence the final performance. For example, if the channel is 3 or 4.

**Weaknesses:**

1. I think the Property (3) in page 4 is very important. And many previous autoregressive models can not solve it well. Actually I think it is not difficult to define an E(3)-equivariant model currently as many previous works have solved it well. The diffusion model doesn't face the permutation-invariant problem. But it can be serious for the autoregressive model. The main paper claims the three properties have been proved in Theorem B.1. Unfortunately, I found the authors only proved the first two properties and ignored the property (3). I DON'T believe the current framework can guarantee the permutation-invariant even the embedder is E(3)-equivariant.

2. I think many criticisms of the diffusion model such as EDM on page 6 are unfair.

a. "Unlike autoregressive models, diffusion models do not flexibly allow for completion of molecular fragments". Actually, it is not difficult for diffusion models to do completion sampling by replacement guidance without any extra training. Currently, the diffusion model works well on image completion and I also try it on EDM, it works too.

b. "To avoid recomputation of the neighbor lists during diffusion, current diffusion models use fully-connected graphs where all atoms interact with each other. This could potentially affect their scalability when building larger molecules." I believe it is even more challenging for the autoregressive model on the large molecules as the sequence could be too long. And the molecules in QM9 used in the experiment are very small. There is no evidence that the proposed framework performs better on large molecules compared with EDM.

c. "diffusion models are significantly slower to sample from, because the underlying neural network is invoked ≈ 1000 times when sampling a single molecule." It is true that for EDM it needs 1000 times denoising process. But actually, it runs very fast and supports batch sampling. I tested EDM on a single GPU A100, it takes 20 seconds to sample 50 molecules (a batch). I think the submission should provide their own sampling time to support this claim.

3. Though the definition of the position distribution is very clear and reasonable, the current submission doesn't provide any details on how to sample from it after training. People can know $f^{position}$ for any given position, but it is not easy to sample from an energy function. I doubt this sampling process can be very time-consuming.

**Questions:**

1. Why did you choose 64 uniformly spaced values from 0.9A to 2.1A? I think the model can learn the distribution automatically by equation (3).

2. Since there are 64 spaced values for $r$, I am wondering how you solve them during the sampling process.

3. Can you give some explanation as to why the EDM outperforms the proposed framework for many metrics?

---

> ### Author Response · Authors · 2023-11-18
> **Response to Reviewer XBL9**
>
> Thank you for your great feedback and taking the time to review our paper! We answer specific questions below. Please also take a look at our common response.
>
> > it lacks an ablation study to show how the multiple channels influence the final performance. For example, if the channel is 3 or 4.
>
> This has now been added in Section F.2 (synthetic task) and Section G.1 (QM9). There is a clear benefit when adding channels for smaller $l$.
>
> > Unfortunately, I found the authors only proved the first two properties and ignored the property (3). I DON'T believe the current framework can guarantee the permutation-invariant even the embedder is E(3)-equivariant.
>
> Sorry for omitting this in our original manuscript. In fact, we had an error in our property (3): we meant to say that the focus distribution should be permutation-equivariant. We have added a proof of this in Theorem B.2. Note that we are not claiming that the entire generation process is invariant to the choice of ordering of nodes; this is almost impossible to guarantee for an autoregressive model. Note that G-SchNet and G-SphereNet both impose some sort of ordering on the way atoms are added. We are also planning to explore some alternate strategies compared to the nearest-neighbours logic in our paper.
>
> > There is no evidence that the proposed framework performs better on large molecules compared with EDM.
>
> This is a fair criticism; we will plan to demonstrate Symphony on the larger GEOM-DRUGS dataset that has been used in other work.
>
> > I think the submission should provide their own sampling time to support this claim.
>
> We have provided the sampling times in our common response above (and in Section 4.4 of the paper). Symphony is currently 3x faster than EDM, but we believe the sampling time for Symphony can be improved by avoiding some of JAX's limitations. We measured the inference times ourselves on the same NVIDIA RTX A5000 GPU.
>
> > Though the definition of the position distribution is very clear and reasonable, the current submission doesn't provide any details on how to sample from it after training.
>
> Sorry for omitting this, we have added this in our common response (and in Section D of the Appendix of the paper). Hopefully this clears up any confusion.
>
> > Why did you choose 64 uniformly spaced values from 0.9A to 2.1A? I think the model can learn the distribution automatically by equation (3).Since there are 64 spaced values for $r$, I am wondering how you solve them during the sampling process.
>
> This should now be clear from Section D of the Appendix. Basically, we do not have a orthonormal basis over the space $(r, \\theta, \\phi)$,  only the space $(\\theta, \\phi)$. Discretization of $r$ was the simplest alternative; we plan to explore other basis functions (and normalizing flows) soon.
>
> > Can you give some explanation as to why the EDM outperforms the proposed framework for many metrics?
>
> Great question! We are also not super sure; the radial discretization that Symphony uses seems to hurt the bond length metrics. However, note that EDM is also trained for 10x longer than Symphony was. We are exploring if the gap can be bridged with further training.

---

### Official Review · Reviewer_UhEf · 2023-11-06

**Soundness:** 2 fair
**Presentation:** 3 good
**Contribution:** 2 fair
**Rating:** 8
**Confidence:** 3

**Summary:**

The authors propose to generate molecules autoregressively, invariant to Euclidean transformations. First, the authors transform the dataset of molecules into a dataset of sequences, adding one atom per time step. This is then turned into a autoregressive generative problem. Iteratively: a "focus" atom in the past is sampled, then the atomic charge of the next atom, then the position of the next atom relative to the focus atom. By having the focus selection and atomic charge distributions invariant to Euclidean transformations, and the position distribution equivariant, the resulting distribution is invariant to Euclidean transformations. The procedure is made permutation invariant by making the target sequences choose the nearest neighbour next, starting from a random atom, and breaking ties randomly.

The network uses higher-order representations of the rotation group to allow for more precision in the position distribution, which is an energy based model / harmonic exponential family on the spherical manifold. In the experiments, the method outperforms other autoregressive methods, while being outperformed by all-on-one diffusion methods.

**Strengths:**

- The code is provided and is readable.
- Autoregressive generation of molecules can be a fast alternative to e.g. diffusion methods.
- The method outperforms other autoregressive methods.
- The translational symmetry is elegantly handled via the focus atom. Also the rotational equivariance relative to the focus atom is sensible, compared to rotations around a center of mass, as is commonly done.

**Weaknesses:**

- in effect, the authors define an energy based model for the positions. It appears from the code that training and sampling of positions is done by discretizing $f^{position}$ on a grid. This seems like a shortcoming of the method, as I suspect that precise positioning is important in molecular generation. The paper should be transparent about this discretization. What type and resolution of grid is used? It'd be interesting to see how the resolution affects the sample quality. If the quality is highest with a very fine grid, it'd be great if the authors could consider and evaluate alternative learning and sampling strategies for energy based models (contrastive divergence / Hamiltonian Monte Carlo / Langevin dynamics etc).
- While the approach predicting the sequence index for the focus point is in/equivariant to permutations of the resulting molecule, it still appears to me that an opportunity for additional permutation equivariance is missing. Why didn't the authors use a permutation equivariant network on $S^n$ to select one focus atom?
- The method performs worse than diffusion methods and don't convince that autoregressive generation is the appropriate method for molecular generation.

When the authors address the gridding issue convincingly, I will raise my score.

**Questions:**

- It appears to me that the position distribution is an instance of a Harmonic Exponential Family [1] on a manifold, which probably should be cited.
- Can the authors clarify why Eq (4) is smooth wrt the radius, but has a Dirac delta in the direction component? This should mean that the KL divergence is infinite almost always? Or is the smoothening handled implicitly by a band-limited Fourier transform? If so, it's not actually a Dirac delta that's used. The authors should clarify that.
- As high frequencies may be necessary to precisely predict positions, it'd be great if the authors can show in an ablation study that $l_{max}=5$ suffices.
- The authors write about $E(3)$ symmetry, which includes mirroring, but subsequently only talk about rotations and translations. Did the authors mean the group $SE(3)$, excluding mirroring?
- Can the authors comment on the runtime of autoregressive generation vs diffusion methods in terms of the number of atoms? Might it be that autoregressive methods are only faster for small molecules?


Refs:
- [1] Cohen, Taco S., and Max Welling. 2015. “Harmonic Exponential Families on Manifolds.” arXiv [stat.ML]. arXiv. http://arxiv.org/abs/1505.04413.

-------
My concerns have been convincingly addressed and I have increased my score.

---

> ### Author Response · Authors · 2023-11-18
> **Response to Reviewer UhEf**
>
> Thank you so much for the great review! We answer specific questions below. Please also take a look at our common response.
>
> > in effect, the authors define an energy based model for the positions.
>
> We have clarified this above; in the autoregressive setting with teacher-forcing, we know the target distribution exactly. Further the partition function can be easily estimated via numerical integration. Thus, we do not need any complicated methods for learning the position distribution, because minimizing KL divergence works well.
>
> > The paper should be transparent about this discretization.
>
> Thank you for mentioning this. As discussed in the common rebuttal, we have added this to Section D of the manuscript. We use a Gaussian Product grid for each value of $r$, which consists of a 1D Gauss-Legendre quadrature with $180$ points over $\\cos \\theta \\in [-1, 1]$, and a uniform grid of $359$ points over $[0, 2\\pi)$ for $\\phi$.
>
> > It'd be interesting to see how the resolution affects the sample quality.
>
> Yes, we have added this in Section G.2 (Figure 13) where we vary the resolution of the grid significantly. The model does not seem particularly unstable to these variations. In Figure 14, we show that the training is also not significantly affected by the resolution for a synthetic task of learning a single distribution on a sphere. We hope that these sets of experiments help alleviate your concern about the gridding procedure.
>
> >  Why didn't the authors use a permutation equivariant network on $\mathcal{S}^n$ to select one focus atom?
>
> We are very sorry for the confusion. We actually do use a permutation equivariant network (GNN) on $\mathcal{S}^n$ to select the focus atom. Our Property (3) incorrectly stated that the focus distribution should be permutation-invariant; this has now been fixed to read permutation-equivariant. We also added Theorem B.2 in the Appendix to prove that this must be the case.
>
> > It appears to me that the position distribution is an instance of a Harmonic Exponential Family [1] on a manifold, which probably should be cited.
>
> Thanks for sharing this relevant article. We have added this as a citation.
>
> > Can the authors clarify why Eq (4) is smooth wrt the radius, but has a Dirac delta in the direction component? This should mean that the KL divergence is infinite almost always? Or is the smoothening handled implicitly by a band-limited Fourier transform?
>
> Yes, you are correct. This is an approximate Dirac Delta distribution that this constructed using the same spherical harmonic projection idea. We have a description of this in Appendix H.
>
> > Did the authors mean the group SE(3) excluding mirroring?
>
> We actually meant E(3) itself, because our proofs handle the case of improper rotations. Our predicted distributions will invert as well under inversions about the origin (mirroring).
>
> > Can the authors comment on the runtime of autoregressive generation vs diffusion methods in terms of the number of atoms? Might it be that autoregressive methods are only faster for small molecules?
>
> We are currently trying to measure this (atleast approximately). The scaling time of EDM seems to be quadratic in the number of atoms (due to fully connected graphs). We will have an update soon.
>
> Thanks again!
>
>
> > As high frequencies may be necessary to precisely predict positions, it'd be great if the authors can show in an ablation study that $l_\\max = 5$ suffices.
>
> Yes, we have added an experiment with Symphony variants trained for different values of $l_\\max$ and number of position channels in Section G.1 (Figure 12)

---

> ### Author Response · Authors · 2023-11-21
> **Follow-up to Edits**
>
> We have edited our response above and the appendix with the changes you recommended. We hope that our additional experiments in Section G.2 help resolve your worries about the discretization. Please let us know if there are any more experiments you would like to see. Otherwise, we would greatly appreciate it if you could improve your score for our paper. Thank you again!

---

> > ### Comment · Reviewer_UhEf · 2023-11-22
> >
> > My concerns have been convincingly addressed and I have increased my score.

---

> ### Comment · Reviewer_UhEf · 2023-12-04
>
> I forgot to add: please consider adding a reference to [1] in a future version. Your way of defining and learning a distribution appears closely related to theirs.
>
> [1] Murphy et al, Implicit-pdf: Nonparametric representation of probability distributions on the rotation manifold, 2021

---

### Public Comment · ~Niklas_Wolf_Andreas_Gebauer1 · 2023-11-13
**Note about related work**

Dear authors,

thank you for this great contribution! I read your manuscript and think that it presents a promising approach and contains thorough, insightful experiments and evaluations.

However, I need to note that spherical harmonics have been used in autoregressive molecule generation before to express the position of the new atom [1]. While the aforementioned method is trained with reinforcement learning, the formulation of the spherical distribution, the generation process, and the overall network used are very similar.

I think it is mandatory to discuss the similarities and differences between your work and [1]. For example, [1] elaborate on difficulties in normalizing the distribution over the sphere and how to sample from it. In your case, the explanation of normalization/sampling is lacking details. Are you using a grid on the sphere for this purpose? If so, how is this grid defined? I think  a comparison/more thorough discussion of this aspect would be very valuable.

Kind regards,
Niklas

[1] Simm, G. N. C., Pinsler, R. Csányi, G. & Hernández-Lobato, J. M. "Symmetry-aware actor-critic for 3d molecular design". In International Conference on Learning Representations, https://openreview.net/forum?id=jEYKjPE1xYN (2021).

---

> ### Author Response · Authors · 2023-11-13
> **Reply: Note about related work**
>
> Thank you for your bringing our attention to this paper! We were unaware of their approach. We will update our paper soon to discuss the differences between Symphony and Simm, et al. (2021) in model design, training and sampling.

---

> > ### Author Response · Authors · 2023-11-18
> > **Follow-up!**
> >
> > We have added a description of our parametrization in Section D, with a comparison to Simm, et al (2021) via a small experiment in Section F.  Thank you again!

---

### Author Response · Authors · 2023-11-18
**Common Response: Part 1**

First, we thank all of the reviewers for their thoughtful feedback. We are glad that Reviewers *UhEf* and *msmy* found Symphony more precise and flexible for modelling the position distribution by utilizing higher-order E(3) equivariant features. Reviewers *XBL9* and *cp6D* appreciated the novelty of spherical harmonic projections with multiple channels. Reviewer *msmy* highlighted that Symphony "boasts a significant advantage in computational efficiency, as current diffusion models rely on fully-connected graphs that may pose scalability challenges".

We first clarify how we represent position distributions. Then, we detail some new experiments and ablation studies for the design choices in Symphony, as recommended by the reviewers. These experiments with their descriptions have been added to Section F and G of the Appendix in our updated manuscript.


## Representing Position Distributions

An aspect of Symphony that was unclear in our original manuscript was how position distributions were learned and sampled. While we do indeed model the logits $f(r, \\theta, \\phi)$, we also have access to the true distribution $q(r, \\theta, \\phi)$. This means we can directly optimize $p$ to match $q$ via the KL divergence:
$$
\def\r{\vec{\mathbf{r}}}
\def\lmax{l_{\text{max}}}
KL(q \\ || \\ p) = \\int_\\Omega q(\\r) \\log \\frac{q(\\r)} {p(\\r)} d\\r =  \\int_\\Omega q(\\r) \\log q(\\r) d\\r - \\int_\\Omega q(\\r) f(\\r) d\\r +\\log Z
$$ where $p$ is obtained by:
$
p(\\r) = \\frac{1}{Z}\\exp f(\\r)
$ and $Z = \\int_\\Omega \\exp f(\\r) d\\r$ is the partition function. $\\Omega$ here represents the sphere, which we represent in spherical coordinates.

**Training**:
For training, we only need the unnormalized logits $f$ and not the normalized distribution $p$. This is identical to the log-sum-exp trick when training with cross-entropy loss for a classification problem. Unlike the classification case where the number of classes is finite, the integral above must be computed over all of $r$, $\\theta$ and $\\phi$ which is an infinite set. To numerically approximate this integral, we use a uniform grid on $r$ and a Spherical Gauss-Legendre quadrature on the sphere at each value of $r$. As discussed in the original submission, the uniform grid on $r$ spans $64$ values from $0.9$A to $2.0$A which is more than sufficient to cover all bond lengths in organic molecules. The Spherical Gauss-Legendre quadrature is a product of two quadratures: a 1D Gauss-Legendre quadrature with $180$ points over $\\cos \\theta \\in [-1, 1]$, and a uniform grid of $359$ points over $[0, 2\\pi)$ for $\\phi$.

Note that Symphony does *not* directly predict $f(r, \\theta, \\phi)$ at each value of $(r, \\theta, \\phi)$. Instead, it predicts the coefficients $c_l(r)$ of $f$ which can be used to evaluate $f(r, \\theta, \\phi)$ at any point. This evaluation for a spherical grid of $(\\theta, \\phi)$ values can be done quickly via a Fast Fourier Transform (FFT) that we have implemented in [e3nn-jax](https://https://e3nn-jax.readthedocs.io/en/latest/api/s2.html#e3nn_jax.to_s2grid). We perform this FFT procedure for each sphere defined by the radius $r$.

With this numerical integration procedure, we can compute the loss only using the logits $f$.

We evaluate the effect of the resolution of the grid on training in Section G.2 (Figure 14).
We see that the KL divergence is well-behaved across a large range of angular grid resolutions. Further, the error in sampling positions is quite low until the grid becomes exceedingly coarse.


**Sampling**:
Once the model is learnt, we need to sample from the distribution represented by coefficients $c_l$. A key advantage of predicting the coefficients $c_l$ is that a different resolution of angular grid can be chosen for sampling than the grid used for training.


## Ablation for Grid Resolution
In Section G.2 (Figure 13), we analyse the validity of molecules generated by our model as a function of the resolution of the grid. We see that the validity is not affected even when using grids $10\\times$ smaller than what the model was trained with. We also investigate the effects of training with different grid resolutions, and find that the learning dynamics are similar across a wide range of grids.

---

> ### Author Response · Authors · 2023-11-18
> **Common Response: Part 2**
>
> ## Ablation for $\\lmax$ and Position Channels
>
> We trained a Symphony on QM9 varying $\\lmax$ for the focus and position embedder, in Section G.1 (Figure 12). Due to computational constraints, we trained these models for $1,000,000$ steps each, which is $8\\times$ lesser than the original model reported in Section 3.3.
>
> For the focus embedder E3SchNet, we did not see a significant increase in validity when going from $\\lmax = 1$ to $\\lmax = 2$, so we kept $\\lmax = 2$.
>
> For the position embedder NequIP, we find a large jump when going from $\\lmax = 1$ to $\\lmax = 2$. Further increasing $\\lmax$ seemed to help slightly. For computational reasons, we kept $\\lmax = 5$.
>
> Note that using our novel multiple channels trick, we are able to represent signals of higher angular frequency with a lower $\\lmax$. This seems to help for $\\lmax = 1$ in particular. However, we believe that the average number of neighbors in QM9 is too low for additional position channels to help at higher $\\lmax$.
>
> To better understand the effect of adding multiple position channels, we explore the synthetic task of  learning a random distribution on the sphere in Section F.2 (Figures 10 and 11). Here, we see that adding multiple position channels significantly helps the expressivity of the model. In particular, a $\\lmax = 4$ signal can be represented with $4$ channels of $\\lmax = 2$. As the computational complexity of many E(3)-equivariant models scales super-linearly with increasing $\\lmax$, we believe the use of multiple channels to represent higher-order signals may be useful beyond autoregressive generation in Symphony.
>
>
>
> ## Training Times with Different Radial Discretizations
>
> Reviewer *msmy* asked us about the effect of radial discretization on the training speed of the model. To investigate this, we changed the number of radii in the radial discretization, keeping all other hyperparameters the same. The following is the approximate training speed of the model for a batch size of $16$ graphs, as measured on a single NVIDIA RTX A5000 GPU:
>
>
>
> | Number of Radii| $32$ | $64$ | $128$ | $256$ |
> | -------- | -------- | -------- | -------- | -------- |
> |  Approximate Training Speed (steps/second)     |   $16.5$   | $13.8$ | $10.1$ | $6.6$ |
>
>
>
> ## Training Times with Different Angular Discretizations
>
> Reviewer *UhEf* asked us about the effect of angular discretization on the training speed of the model. To investigate this, we changed the discretization of $\\phi$, keeping all other hyperparameters the same. $r_\\theta$ was kept as $180$ as in the original model. The following is the approximate training speed of the model for a batch size of $16$ graphs, as measured on a single NVIDIA RTX A5000 GPU:
>
>
> | $r_\\phi$ |  $49$ | $89$ | $179$ | $359$ | $719$
> | -------- | -------- | -------- | -------- | -------- | -------- |
> |  Approximate Training Speed (steps/second)     |   $20.5$   | $19.1$ | $16.5$ | $14.1$ | $10.2$
>
> We do see an increase in the training speed, but not a linear one.
>
> ## Sampling Times
>
> Reviewers *XBL9* and *cp6D* requested further details on the sampling time for Symphony compared to other models. These were present in Section 4.4 of the original manuscript. We restate these numbers here, as measured on a single NVIDIA RTX A5000 GPU:
>
>
> | | Symphony | EDM | G-SchNet | G-SphereNet |
> | -------- | -------- | -------- | -------- | -------- |
> |  Sampling Time (secs/mol)     |   $0.293$   | $0.930$     | $0.011$ | $0.006$ |

---

> ### Author Response · Authors · 2023-11-18
> **Common Response: Part 3**
>
> ## Comparison to Simm, et al. (2021)
>
> We greatly appreciate the [public comment](https://openreview.net/forum?id=MIEnYtlGyv&noteId=jUiATOXVJR) by Niklas drawing our attention to Simm, et al. (2021) [1] which also uses spherical harmonic projections molecule generation with reinforcement learning. We were not aware of this work during the preparation of this manuscript. Here, we highlight some differences between Symphony and their model:
> * Given coefficients $c_l$, Symphony constructs the probability distribution $p$:
> $$
> \\begin{aligned}
> f(r, \\theta, \\phi) &= \\sum_{l = 0}^{\\lmax} c_l(r)^T Y_l(\\theta, \\phi)  \\\\
> p(r, \\theta, \\phi) &= \\frac{1}{Z}\\exp f(r, \\theta, \\phi)
> \\end{aligned}
> $$ while [1] constructs the probability distribution $p$ as a product of radial density $p_r$ which is a mixture of $M$ Gaussians and an angular density $p_{\\theta, \\phi}$ conditioned on $r$:
> $$
> \\begin{aligned}
> p_r(r) &= \\sum \\pi_m \\mathcal{N(\\mu_m, \\sigma_m^2)} \\\\
> p(\\theta, \\phi \\ | \\ r) &= \\frac{1}{Z}\\exp \\left(-\\frac{\\beta}{k} |f(r, \\theta, \\phi)|^2  \\right ) \\\\
> p(r, \\theta, \\phi) &= p_r(r)p_{\\theta, \\phi}(\\theta, \\phi \\ | \\ r)
> \\end{aligned}
> $$ where $k = {\\sum_{l = 0}^{\\lmax} |c_l|^2}$. $Z$ in both cases refers to the partition constant which normalizes the distributions. The extra factor of $k$ in [1] "regularizes the distribution so that it does not approach a delta function". However, in our case, we want to exactly learn a delta distribution (or a mixture of them). We show that their regularization is not ideal for our setting in Section F.2 (Figure 10 and 11).
> * Our use of multiple channels allows representing signals of higher angular frequency than possible with the architecture in [1] as demonstrated by our experiments in "Ablation for Position Channels". [1] also noted the relation between the peakedness of the distribution and $\\lmax$, which our multiple channels trick helps with.
> * To sample from the position distribution, [1] uses rejection sampling with a uniform base distribution. Our sampling strategy is much more efficient; we normalize the logits across the grid points with a softmax, and then simply sample from the grid points as a categorical distribution.
> * The respective tasks are quite different; [1] provides the model with a "bag" of atom types with respective counts, while Symphony learns the distribution of atom types and counts. Further, due to the reinforcement learning setup, they have to sample many times from their model distribution to learn to optimize the true reward. In our case, we can directly learn the model distribution with the KL-divergence relative to the target distribution, without any sampling. This is why Symphony is able to learn much more efficiently; the model in [1] requires over $40,000$ sampling steps to learn the geometry of a single molecule.
> * Symphony is trained and evaluated against the QM9 dataset of $\\approx 130,000$ molecules. Due to computational costs, [1] only tests their model on $9$ molecules. This makes it particularly hard to compare the performance differences at scale.
>
> References:
> * [1]: Simm, G. N. C., Pinsler, R. Csányi, G. & Hernández-Lobato, J. M. "Symmetry-Aware Actor-Critic for 3D Molecular Design". In International Conference on Learning Representations, https://openreview.net/forum?id=jEYKjPE1xYN (2021).
>
>
> ## Other Changes
>
> We found a bug in our evaluation of G-SchNet for the experiments in Section 4.4. These numbers have been updated. We have also added results for Symphony at later checkpoints; we are investigating why the fragment completion numbers drop while the unconditional generation performance increases. We also updated Figure 3 for clarity.

---

> ### Comment · Reviewer_UhEf · 2023-11-20
> **Positional distribution**
>
> I thank the authors for their response. I have some follow-up questions, mostly regarding your wording.
>
> - An energy-based model is typically [1] defined as exactly as a function $f$ such that $p(x)\propto \exp f(x)$, without having an explicit model for $p$, so you're definitely learning an energy-based-model. Could you clarify? It might be that you're confusing the term "energy-based models" this with the contrastive divergence sampling-based training methods for energy based models, but that's only one specific training way of training energy based models.
> - You write that you don't need the normalized distribution for your training objective and write "This also avoids the key issue of estimating the partition function that occurs with energy-based models.", but you do need to estimate Z, as it depends on the parameters of $f$. So you are computing the normalized $\log p$ for your objective, estimated via integration, correct? Of course, as your space is only 3D, using an energy-based-model and integrating is a feasible strategy, different from higher-dimensional problems. I think the wording in your reply and revised appendix is not accurate, though.
> - You write that you have access to the true distribution $q$. Do you mean that you know the density? Or do you mean that you have access to samples? Could you clarify?
> - If you're integrating over 4 million points for each sample during training, how does this affect training times?
> - Finally, do you agree that the method introduced in sec 3.4 can be interpreted as a mixture of energy based models? The mixture coefficients are proportional to the partition function for each channel.
>
> Thanks!
>
>
> [1] Teh, Yee Whye, Max Welling, Simon Osindero, and Geoffrey E. Hinton. 2003. “Energy-Based Models for Sparse Overcomplete Representations.” JMLR

---

> > ### Author Response · Authors · 2023-11-20
> > **Response: Positional distribution**
> >
> > Thank you again for your response!
> >
> > > Could you clarify? It might be that you're confusing the term "energy-based models" this with the contrastive divergence sampling-based training methods for energy based models, but that's only one specific training way of training energy based models.
> >
> > Ah, yes. You are indeed correct. With this definition, we are indeed training an energy-based model defined by the spherical harmonic projection coefficients predicted by a neural network. We do not use contrastive divergence sampling-based training methods, and instead optimize the model by KL divergence minimization (maximum likelihood estimation). As part of the optimization,  $\log Z$ is estimated by numerical integration over spheres.
> >
> > > but you do need to estimate Z, as it depends on the parameters of f.
> >
> > Yes, correct, as explained above. But we don't need to explicitly compute $p(r, \theta, \phi)$ for our loss, as explained before. We only need $f(r, \theta, \phi)$ and $\log Z$. (Of course, $p(r, \theta, \phi)$ can be computed from these quantities).
> > This helps with numerical stability.
> >
> > > You write that you have access to the true distribution.
> >
> > The true density (Equation 7 in Section 3.5) is defined as a smooth approximation to a Dirac delta distribution centered at the target sample. This is similar (but not identical) to that of G-SchNet.
> >
> > > If you're integrating over 4 million points for each sample during training, how does this affect training times?
> >
> > Good point! In our "Common Response 2", we had originally mentioned the training times as a function of radial discretization.
> > We have also added a table for the training times as a function of angular discretization $r_\\phi$, copied here for your reference:
> >
> > | Number of Radii| $32$ | $64$ | $128$ | $256$ |
> > | -------- | -------- | -------- | -------- | -------- |
> > |  Approximate Training Speed (steps/second)     |   $16.5$   | $13.8$ | $10.1$ | $6.6$ |
> >
> >
> > | $r_\\phi$ |  $49$ | $89$ | $179$ | $359$ | $719$
> > | -------- | -------- | -------- | -------- | -------- | -------- |
> > |  Approximate Training Speed (steps/second)     |   $20.5$   | $19.1$ | $16.5$ | $14.1$ | $10.2$
> >
> > We do see an increase in training speed but not a linear one, indicating that this is not likely to be the bottleneck.
> >
> > > do you agree that the method introduced in sec 3.4 can be interpreted as a mixture of energy based models?
> >
> > Yes, that seems accurate to us.

---

### Meta-Review · Area_Chair_3ayr · 2023-12-09

**Metareview:**

This paper studies molecule generation using message-passing with higher-degree equivariant features. After rebuttals and discussions, all reviewers acknowledge the novelty of this work along with experimental results, which I agree. Thus an accept is recommended.

**Justification For Why Not Higher Score:**

The work is novel, but not completely novel, as molecule generation has been studied for a while, such as G-Schnet, G-SphereNet etc.

**Justification For Why Not Lower Score:**

The work has technical and empirical values.

---

### Decision · Program_Chairs · 2024-01-16

Accept (poster)